# Tensor-structured decomposition improves systems serology analysis

Zhixin Cyrillus Tan[1], Madeleine C Murphy[2], Hakan S Alpay[3], Scott D Taylor[4] &
Aaron S Meyer[1,4,5,6,*]

## Abstract

Systems serology provides a broad view of humoral immunity by profiling both the antigen-binding and Fc properties of antibodies. These studies contain structured biophysical profiling across disease-relevant antigen targets, alongside additional measurements made for single antigens or in an antigen-generic manner. Identifying patterns in these measurements helps guide vaccine and therapeutic antibody development, improve our understanding of diseases, and discover conserved regulatory mechanisms. Here, we report that coupled matrix–tensor factorization (CMTF) can reduce these data into consistent patterns by recognizing the intrinsic structure of these data. We use measurements from two previous studies of HIV- and SARS-CoV-2-infected subjects as examples. CMTF outperforms standard methods like principal components analysis in the extent of data reduction while maintaining equivalent prediction of immune functional responses and disease status. Under CMTF, model interpretation improves through effective data reduction, separation of the Fc and antigen-binding effects, and recognition of consistent patterns across individual measurements. Data reduction also helps make prediction models more replicable. Therefore, we propose that CMTF is an effective general strategy for data exploration in systems serology.

**Keywords** effector function; HIV; SARS-CoV-2; systems serology; tensor decomposition

**Subject Categories** Computational Biology; Immunology; Microbiology, Virology & Host Pathogen Interaction

**Mol Syst Biol. (2021) 17: e10243**

## Introduction

Whether during a natural infection, therapeutic vaccination, or an exogenously administered antibody therapy, antibody-mediated protection is a central component of the immune system. The unique property of antibodies is conceptually simple—they undergo affinity enrichment toward specific antigens—but the mechanisms of resulting protection are mediated through a network of interactions (Lu *et al*, 2017). Therapies are often optimized based upon the titer or neutralizing capacity of the antibodies they deliver. However, many of the mechanisms for antibody-mediated protection occur through secondary interactions with the immune system via an antibody's fragment-crystallizable (Fc) region. Although more challenging to quantify and identify as the mechanism of protective immunity, these immune system responses, such as antibody-dependent cellular cytotoxicity (ADCC) (Hessell *et al*, 2007; Bournazos *et al*, 2014), complement deposition (ADCD) (Lofano *et al*, 2018), cellular phagocytosis (ADCP) (Osier *et al*, 2014), and respiratory burst (Joos *et al*, 2010), are known to be just as or more important in many diseases.

A suite of recent technologies promises to broaden our view of antibody-mediated protection as the microarray did for gene expression. Systems serology aims to broadly profile the humoral immune response by jointly quantifying both the antigen-binding and Fc biophysical properties of antibodies in parallel (Arnold & Chung, 2017; Chung & Alter, 2017). In these assays, antibodies are first separated based on their binding to a panel of disease-relevant antigens (Brown *et al*, 2012, 2017). Next, the binding of the immobilized antibodies to a panel of immune receptors is quantified. Other molecular properties of the disease-specific antibody fraction that affect immune engagement, such as glycosylation, may be quantified in parallel in an antigen-specific or -generic manner (Brown *et al*, 2012, 2017; Mahan *et al*, 2014). By accounting for the two necessary events for effector response—antigen binding and immune receptor engagement—these measurements have proven to be highly predictive of effector cell-elicited responses and overall

1 Bioinformatics Interdepartmental Program, University of California, Los Angeles, Los Angeles, CA, USA
2 Computational and Systems Biology, University of California, Los Angeles, Los Angeles, CA, USA
3 Department of Computer Science, University of California, Los Angeles, Los Angeles, CA, USA
4 Department of Bioengineering, University of California, Los Angeles, Los Angeles, CA, USA
5 Jonsson Comprehensive Cancer Center, University of California, Los Angeles, Los Angeles, CA, USA
6 Eli and Edythe Broad Center of Regenerative Medicine and Stem Cell Research, University of California, Los Angeles, Los Angeles, CA, USA
*Corresponding author. Tel: ++1 310 794 4821; E-mail: a@asmlab.org

antibody-elicited immune protection (Chung *et al*, 2015; Alter *et al*, 2018; Zohar *et al*, 2020).

Although systems serology provides a major advancement in our ability to analyze the antibody-elicited immune response, analysis of these data is often challenging. Standard machine learning methods, such as regularized regression, principal components analysis (PCA), and partial least squares regression (PLSR), have been effective in identifying highly predictive immune correlates of protection (Choi *et al*, 2015; Alter *et al*, 2018). However, identifying specific molecular changes or programs that give rise to protection is more difficult. First, because many of the measurements are overlapping in the molecules they quantify, or measure co-dependent processes, much of the data are highly intercorrelated (Chung *et al*, 2015; Pittala *et al*, 2019). Particularly when analyzing polyclonal antibody responses such as those which arise in vaccination or natural infection, protection may arise through single or combinations of molecular species and features within the antibody response, through either individual or combinations of antigens (Ackerman *et al*, 2016, 2018). One successful approach in serology analysis has been to collapse molecular features into summary statistics, such as Fc breadth or polyfunctionality, although this requires predefined descriptors of these quantities (Ackerman *et al*, 2016). Alternatively, patterns of interest can be experimentally derived, such as with blocking experiments, but this is labor-intensive and requires pre-existing monoclonal antibodies to define each pattern (Georgiev *et al*, 2013). Unsupervised approaches that explicitly integrate patterns across both antigen and Fc properties will help mechanistically characterize immune protection.

While systems serology measurements include a variety of different assays to quantify humoral response, a common overall structure exists to the data. Most of the measurements quantify the extent to which an antibody bridges all pairs of target antigen and receptor panels, across a set of individuals (Arnold & Chung, 2017). Binding to the target antigen involves the antigen-binding fragment (Fab) of an antibody, whereas immune receptor interactions occur through its Fc region. Thus, it is natural to split them up as they entail different regulatory processes. Along with the dimension of individuals, these measurements, therefore, can be thought of as a three-dimensional dataset, where every number in this "cube" of data represents a single measurement (Fig 1A and B). Then, separately from these measurements, some properties of the humoral response, such as antibody glycosylation, may be assessed but without separation across different antigens (Ackerman *et al*, 2013; Mahan *et al*, 2014). With data of three or more dimensions, tensor decompositions, a family of unsupervised dimensionality reduction methods for higher order tensors, provide a generalization of matrix decomposition techniques (Kolda & Bader, 2009). These methods are especially effective at data reduction when measurements have meaningful multidimensional features, such as time-course measurements (Martino *et al*, 2020). Like PCA, tensor decomposition methods, when appropriately matched to the structure of data, help visualize its variation, reduce noise, impute missing values, and reduce dimensionality (Omberg *et al*, 2007).

As the structure of systems serology data is well-suited to tensor decomposition, we take advantage of this to reap the aforementioned benefits. As examples, we analyze two separate studies wherein systems serology measurements were shown to predict both functional immune responses and disease status within HIV- and SARS-CoV-2-infected subjects (Alter *et al*, 2018; Data ref.: Zohar *et al*, 2020). We first adapt a tensor decomposition approach—coupled matrix–tensor factorization (CMTF)—to reduce these measurements into consistent patterns across subjects, immunologic features, and antigen targets. Inspecting these factors reveals interpretable patterns in the humoral response, and these patterns' abundance across subjects predicts subjects' functional immune responses and infection state. Importantly, CMTF greatly improves the interpretability of these predictions compared with methods that do not recognize the structure of these data. This approach, therefore, provides a very general data-driven strategy for improving systems serology analysis.

# Results

## Systems serology measurements can be arranged in tensor form for greater dimensionality reduction

We first sought to determine whether the structure of systems serology measurements could inform better data reduction strategies (Fig 1). As an array-based measurement, wherein the amount of signal is dependent upon the quantity of both antigen and Fc interactions, we surmised that upon arranging measurements according to the antigen or Fc feature assessed, we might more effectively identify patterns within the data (see detailed justification in Methods). We started by restructuring the HIV infection serology data (Alter *et al*, 2018). To integrate the antigen-specific array and gp120-exclusive glycan measurements, we used a form of tensor-based dimensionality reduction, coupled matrix–tensor factorization (CMTF; Fig 1B and C). By concatenating both the unfolded tensor and matrix during the alternating least squares (ALS) solving for the subject dimension, we achieve the optimal low-rank approximation for both datasets (Fig 1D, see Materials and Methods). This structure is like canonical polyadic (CP) decomposition on a single tensor, or PCA on a single matrix (Fig 1D). The approximation aims to explain the maximal variance across both datasets, in contrast to partial least squares regression in the matrix or tensor form (tPLS), which would explain only the shared variance (Fig 1D).

To determine the extent of data reduction possible, we examined the reconstruction error upon decomposition with varying numbers of components (Fig 2A). As the datasets were formatted into a three-mode (i.e., axis) $181 \times 22 \times 39$ tensor and a $181 \times 25$ matrix, we start with a structure of 159,823 entries, of which 95,484 or ~ 60% were filled with measurements (Fig 1B). After factorization with 6 components, we are left with four matrices of $181 \times 6$, $22 \times 6$, $39 \times 6$, and $25 \times 6$. Therefore, we reduce the dataset to ~ 1.7% of the size ($[181 + 22 + 39 + 25] \times 6 = 1{,}602$ numbers), while preserving 62% of its variation (Fig 2A). For comparison, Fc array assays where these measurements came from with sufficient dynamic range reproduce roughly 80% of the variance across replicates (Brown *et al*, 2012). Therefore, we are capturing the majority but not quite all true variation across subjects and measurements. We compared this with the data reduction possible with PCA with the data organized in a flattened matrix form. CMTF consistently led to a similar variance explained with half the resulting factorization

                                                          

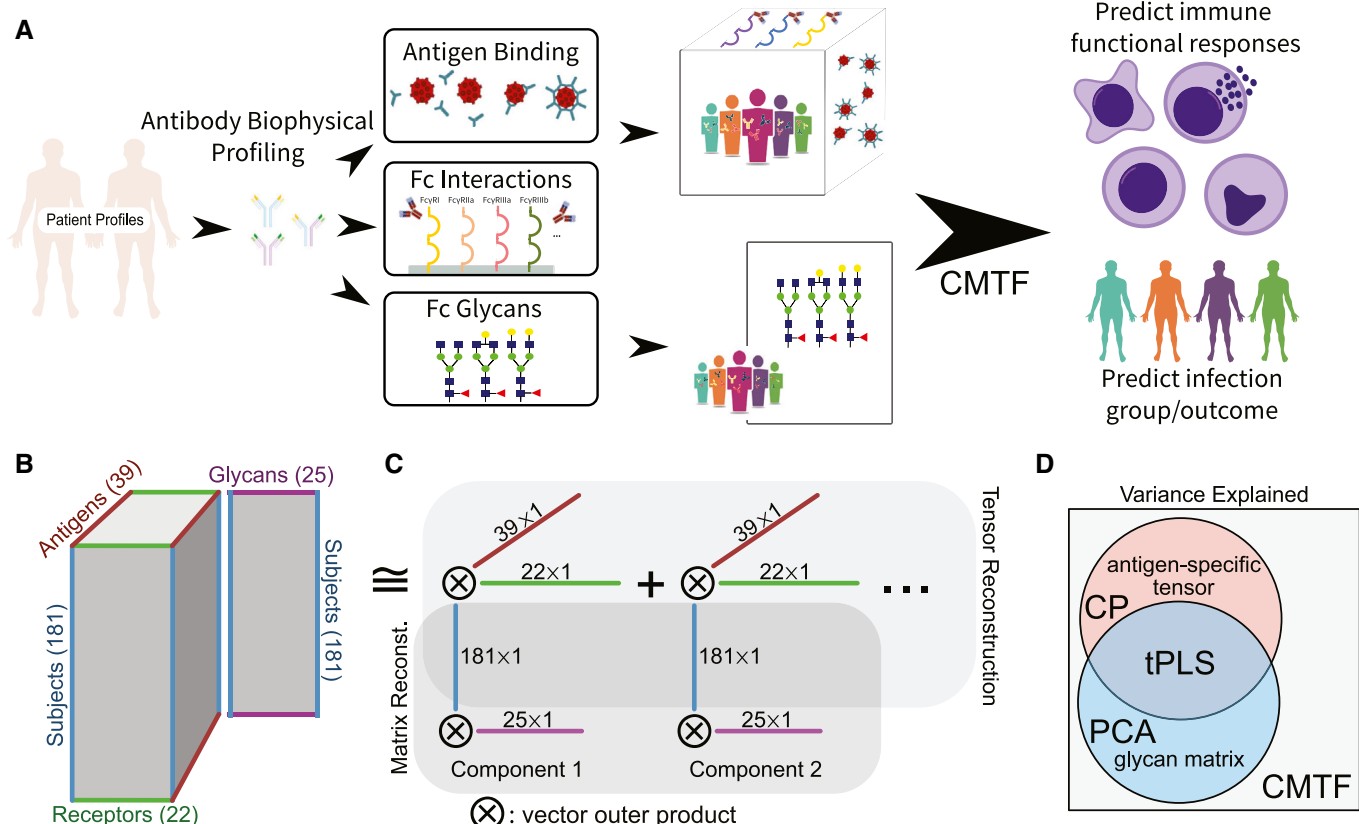

**Figure 1. Systems serology measurements have a consistent multimodal structure.**

A  General description of the data. Antibodies are first separated based on their binding to a panel of disease-relevant antigens. Next, the binding of those immobilized antibodies to a panel of immune receptors is quantified. Other molecular properties of the disease-specific antibody fraction that affect immune engagement, such as glycosylation, may be quantified in parallel in an antigen-specific or -generic manner. These measurements have been shown to predict both disease status (see methods) and immune functional properties—ADCD, ADCC, antibody-dependent neutrophil phagocytosis (ADNP), and natural killer cell activation measured by IFNγ, CD107a, and MIP1β expression.

B  Overall structure of the data under the CMTF framework. Antigen-specific measurements can be arranged in a three-dimensional tensor wherein one dimension each indicates subject, antigen, and receptor. In parallel, non-antigen-resolved measurements such as quantification of glycan composition can be arranged in a matrix with each subject along one dimension, and each glycan feature along the other. Although the tensor and matrix differ in their dimensionality, they share a common subject dimension.

C  The data are reduced by identifying additively separable components represented by the outer product of vectors along each dimension. The subject dimension is shared across both the tensor and matrix reconstruction.

D  Venn diagram of the variance explained by each factorization method. Canonical polyadic (CP) decomposition can explain the variation present within the tensor on its own (Omberg *et al*, 2007), analogous to principal component analysis (PCA) on the glycan matrix. Tensor partial least squares regression (tPLS) allows one to explain the shared covariation between the matrix and tensor (Zhang & Li, 2017). In contrast, here, we wish to explain the total variation across both the tensor and matrix (Choi *et al*, 2019). This is accomplished with CMTF (see Materials and Methods).

size compared with PCA (Fig 2B). For example, as indicated by the arrow, CMTF led to a normalized unexplained variance of 0.45 at ~ 1,024 values within the factorization, whereas PCA required ~ 2,048 to do the same. The difference between PCA and CMTF must arise from the latter's ability to "reuse" antigen patterns across receptors, or vice versa. For example, if a component includes an increase in FcγRIII binding overall, PCA would still need to represent this increase in the loadings for every FcγRIII-antigen measurement. Thus, PCA is not able to "group" interaction effects across the two dimensions. The difference cannot arise through relaxing orthogonality; CMTF is still "hyper-orthogonal" (i.e., full rank across all tensor modes), and linearly dependent components would only reduce the total variance explained (Kolda & Bader, 2009). Overall, highly effective dimensionality reduction gave us

confidence that this structured factorization identifies patterns of meaningful variation.

As CMTF aims to maximize the explained variances across both datasets, their relative scale influences the balance of the decomposition (Fig 2C). We standardized the data during preprocessing by scaling the matrix so that it contains the same amount of variance as the tensor. In this case, CMTF explains ~ 62% of the tensor and ~ 40% of the matrix variance (R2X). When the matrix is scaled to relatively larger variance, CMTF can achieve ~ 72% matrix R2X, at the expense of the tensor R2X dropping below 35%. Conversely, a smaller matrix does not increase the tensor R2X over 65% but causes the matrix R2X to decrease sharply. Our approach of equal variance scaling tuned the factorization to the range where it was responsive to both datasets.

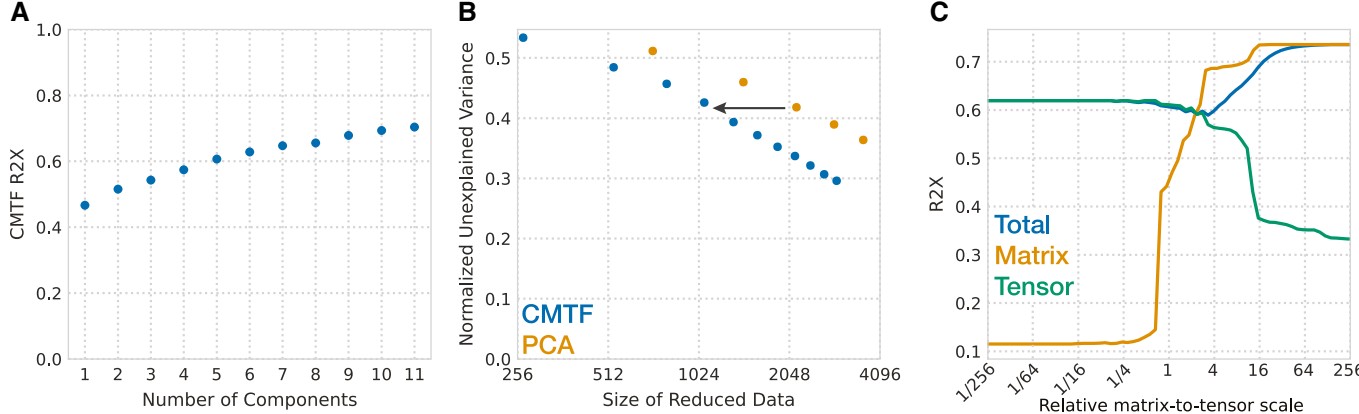

**Figure 2. CMTF improves data reduction of systems serology measurements.**

A   Percent variance reconstructed (R2X) versus the number of components used in CMTF decomposition.
B   CMTF reconstruction error compared with PCA over varying sizes of the resulting factorization. The unexplained variance is normalized to the starting variance. Note the log scale on the *x*-axis. CMTF consistently led to a similar variance explained with half the resulting factorization size compared with PCA. For example, as indicated by the arrow, to obtain a normalized unexplained variance of 0.45, PCA required ~ 2,048 values, and CMTF needed only ~ 1,024 values.
C   The overall and matrix- or tensor-specific R2X with varied relative scaling.

## Factorization accurately imputes missing values

By rearranging the measurements into a tensor form, our data structure created an entry for every combination of antigen, subject, and Fc property. However, as not every quantity represented by these entries was measured in the dataset, this tensor was left with empty positions or missing values. To demonstrate that CMTF was robust with missing values, we benchmarked its ability to impute them.

Missing data are not uncommon to biological research. In an experiment, subject samples can be limited or only be available for a small set of measurements, or a subset of measurements can be prioritized by investigators based on prior knowledge. Incapable of handling missing values, one may have to exclude incomplete measurements. In the HIV serology data, gp120-specific glycan measurements were available for only half of the subjects. Consequently, models using the glycan measurements required a smaller patient cohort, and when they were included, the prediction performance reduced (Alter *et al*, 2018). Good imputation performance can not only potentially eliminate such trade-off but also help infer unknown information. Moreover, factorization accurately imputing missing values further supports that this approach identifies biologically meaningful and consistent patterns.

To evaluate the imputation performance of factorization, we first artificially introduced additional missing values by randomly removing *entire* receptor–antigen pairs *across* all subjects (see Materials and Methods). We then performed CMTF which effectively filled these in and calculated the Q2X of the inferred values compared with the left-out data (Fig 3A). Factorization imputed these values with similar accuracy to the variance explained within observed measurements up to six components (Fig 2A), supporting that it can identify meaningful patterns even in the presence of missing measurements. As we were effectively leaving out entire columns of data when arranged in a flattened matrix form, we could not compare this performance with PCA. Using the average along the receptor or antigen dimensions led to Q2X values very close to 0. As

a less stringent imputation task, we left out batches of individual values and evaluated our ability to impute them. CMTF showed similar or slightly better performance when imputing individual values compared with PCA (Fig 3B). This provides additional evidence that the patterns identified by factorization are a meaningful representation of the data.

## Tensor decomposition accurately predicts functional measurements and subject classes

We next evaluated whether our reduced factors could predict the functional responses of immune cells and subject classes. Functional responses included antibody-dependent complement deposition (ADCD); cellular cytotoxicity (ADCC); neutrophil phagocytosis (ADNP); and the level of natural killer (NK) cell activation represented by the expression of IFNγ, CD107a, and MIP1β. Subject classes included whether subjects were able to control their infection and whether they were viremic at the time of study collection.

To predict the functional responses, we applied elastic net to the decomposed factors (see Materials and Methods), and their prediction accuracies were defined as the Pearson correlation between measured and predicted values (Fig 4A and C). To predict the subject classes, we applied logistic regression (see Materials and Methods), and accuracy was defined as the percent classified correctly (Fig 4B and D). To evaluate prediction, we implemented a 10-fold cross-validation strategy. In brief, in each fold, we used 90% of the subjects to learn the relationship between the data and the given prediction and then evaluated these predictions on the remaining 10%. The average performance of each approach was evaluated with every subject eventually held out in one of the 10 folds.

To determine the optimal number of components, we first evaluated the prediction accuracies of CMTF with 1–14 components (Fig 4A and B). With more components, functional response prediction accuracies improved marginally and mostly plateaued after 6

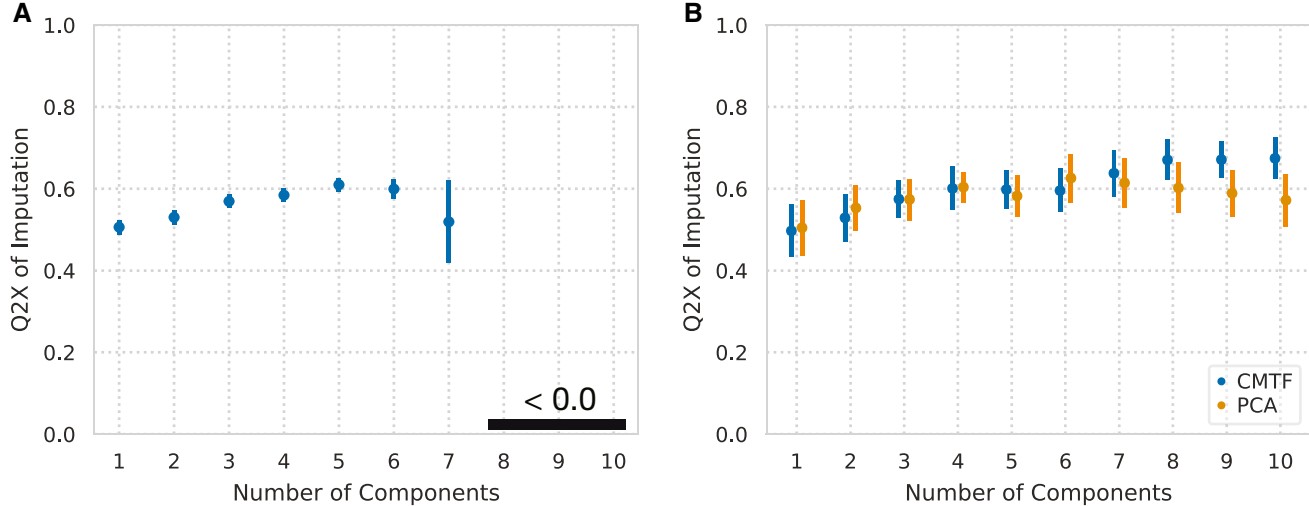

**Figure 3.  CMTF accurately imputes missing values.**

A   Percent variance predicted (Q2X) versus the number of components used for imputation of 15 randomly held-out receptor–antigen pairs. Error bars indicate standard error of the mean from repeatedly held-out pairs (N = 20).
B   Percent variance predicted (Q2X) versus the number of components used for 15 randomly held-out individual values. Error bars indicate standard error of the mean from repeatedly held-out values (N = 10).

components. Subject class predictions saw a leap from 3 to 4 components, especially for controller–progressor classification, and all class prediction accuracies plateaued after 6 components. We therefore concluded that 6 components were generally sufficient for good predictions.

For comparison, we reimplemented the elastic net-based immune functionality and subject predictions previously applied to these data (Fig 4C and D, orange crosses) (Alter *et al*, 2018). We observed similar performance to that reported. Differences from reported results could be explained by adjustments we made to the cross-validation strategy to prevent over-fitting (see methods). Broadly, we saw overall our method performed similarly to the previous method in predicting immune functional responses and subject classes (Fig 4C and D, blue circles). Although lower at 6 components, our prediction accuracy increased slightly for ADCD and ADNP at higher numbers of components (Fig 4A). CMTF also had similar prediction accuracy for subject classes with 6 components (Figs 4D and EV1). Importantly, in all cases, randomizing the subjects' classes completely removed the ability to make these predictions (Fig 4C and D, green squares).

As both functions and subject classes were predicted with linear models, we plotted the component weights for these regression results (Fig 4E and F). All the NK activation measurements (IFNγ, CD107a, and MIP1β) were highly correlated (Pearson correlation > 0.85) and unsurprisingly had very similar model weights, but ADCD, ADCC, and ADNP differed more (Fig 4E). To quantify the stability of these component–function and component–class relationships, we performed bootstrapping by resampling the subjects with replacement and included error bars representing the standard deviation of the model weights (Fig 4E and F). In every case, most of the model weights varied little across samples. By contrast, bootstrapping the elastic net model

of ADCD based on the original measurements themselves, as an example, led to entirely different model weights (Fig EV2). We overall concluded that CMTF preserves sufficient information to predict these important features. Data reduction enables one to identify patterns that are associated with functional responses and subject classes, and component associations are more robust upon resampling.

## Factor components represent consistent patterns in the HIV humoral immune response

We plotted the results of our factorization in four factor plots to inspect the composition of each component across each factor dimension (Fig 5). After ALS, components were ordered by their variance, with component 1 having the greatest variance and component 6 having the least. Because the effect of a component is the product of weights on three modes, the original tensor is invariant to coordinated sign flipping or scaling. We enforced that the receptor and antigen factors are positive on average by cancelling out negative effects along two factor modes. Factor components were also scaled to fall within the range of −1 to 1, and their scaling factors were 29.3, 12.4, 7.4, 7.2, 14.0, and 3.9 respectively. We elected to not scale the glycan factors on a per-component basis so that the relative scaling is evident in the plot itself (Fig 5D). Every component must be distinct along at least one factor matrix due to hyper-orthogonality, so no component was redundant.

The resulting factor plots can be read in two ways. First, one can trace the effect of a component across different dimensions by looking at that component within each plot. For instance, component 4 represents a subset of unique variation in the data that is higher in viremic controllers (Fig 5A), broadly covarying across FcγRs (Fig 5B), and increases p24 or decreases gp120 antigen binding (Fig 5C). In an

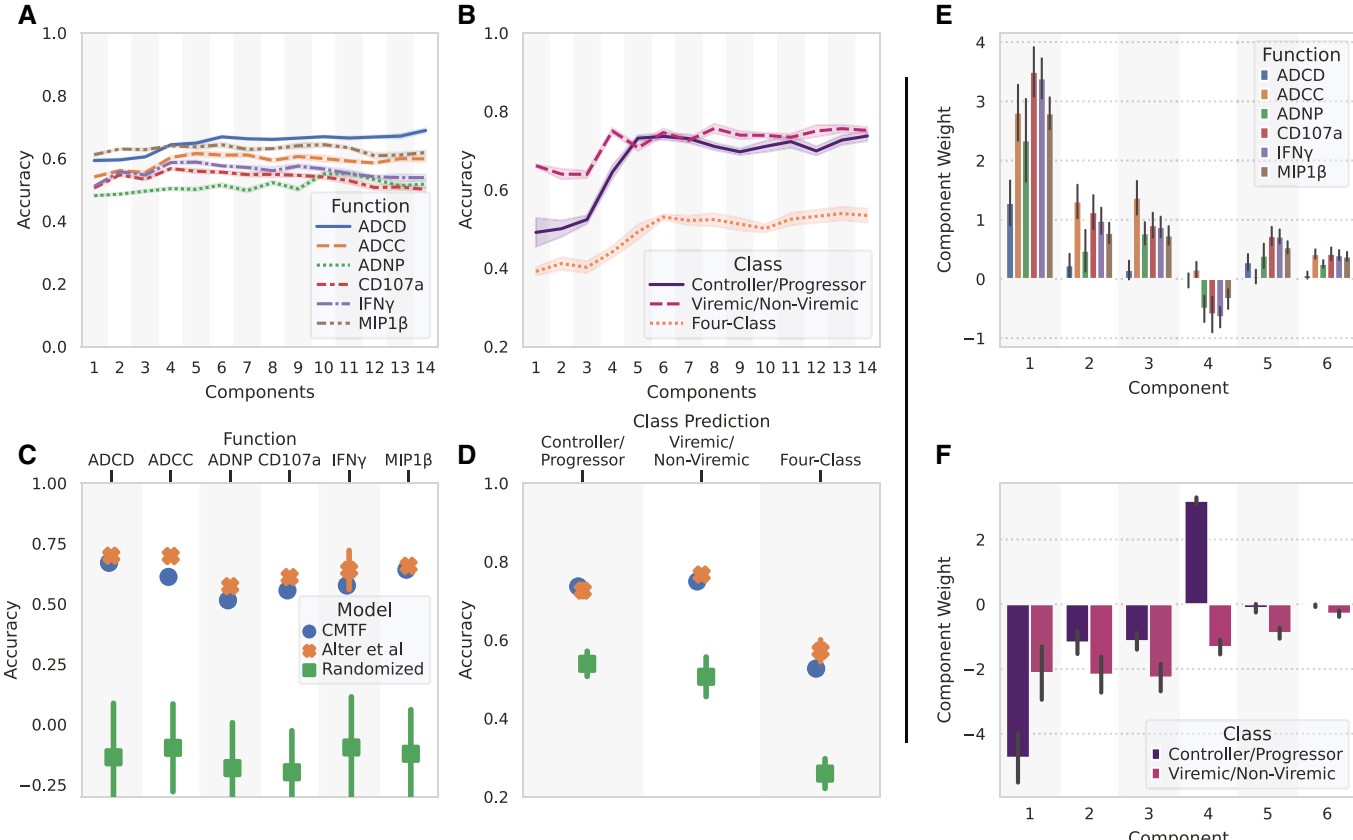

**Figure 4. CMTF-reduced factors accurately predict functional measurements and subject classes.**

A Accuracy (defined as the Pearson correlation coefficient) of functional response predictions with different numbers of components.

B Percent of subject classes predicted accurately with different numbers of components.

C Prediction accuracy for different functional response measurements using six components.

D Fraction predicted correctly for subject viral and controller status using six components.

E, F Model component weights for each function (E) and subject class (F) prediction.

Data information: The shaded area/error bars in (A–D) come from repeating a 10-fold cross-validation (with 10 differently shuffled folds) 10 times ($N = 10$), and the error bars in (E, F) come from bootstrapping 20 times ($N = 20$). All error bars indicate the standard deviation from repeated resampling.

alternative view of the factorization results, one can ask how components are different in the variance they explain within a single factor mode. For instance, components 2 and 4 are very similar in their receptor interactions (Fig 5B) but unique in their antigen binding specificity (Fig 5C). Finally, the product of subject (Fig 5A) and glycan (Fig 5D) factors reconstructs the glycan measurements.

Components 1 and 2 explained the most variance and had broad receptor (Fig 5B) and antigen (Fig 5C) weighting, indicating that they represent overall titers in a general manner. Some difference exists within both components in their antigen specificity—component 1 is weighted toward surface antigens, whereas component 2 is more uniform in its antigen weights (Fig 5C). Component 1 (along with component 4) was also uniquely high in viremic controllers compared with other groups (Fig 5A). Component 3 represents a similar antigen specificity to component 2 (Fig 5C) and similar receptor set, except for most of the lectin-binding proteins (MBL, PNA, SNA, VVL) and C1q (Fig 5B). Component 4 displayed similar receptor specificity to component 1 (Fig 5B) but with unique antigen specificity that was positive for intracellular antigens and negative

for surface ones (Fig 5C). Component 5 was surface antigen-specific (Fig 5C) and strongly specific for LCA, PNA, and VVL (Fig 5B). Finally, component 6 was weighted toward genotype-specific FcγR measurements over all others (Fig 5B), with broad antigen specificity (Fig 5C). As these were (i) the most sensitive measurements as indicated by their generally higher fluorescence signal before normalization and (ii) the component's variation was greatest for the subjects that were low on component 1 (Fig 5A), we took this to indicate the component explained variation specific to low-titer subjects.

We were surprised to find little unique variation in the glycan matrix factor along each component (Fig 5D). The weights within each component were proportional to the dynamic range of each measurement (most for G2S2 and less for total G0 as an example; $r^2 = 0.82$). We took this to indicate that there is little variation explained in the glycan data beyond an overall increase or decrease. As independent evidence of this, a one-component PCA decomposition of just the glycan matrix could explain > 70% of the variation in the glycan data, even after centering.

 

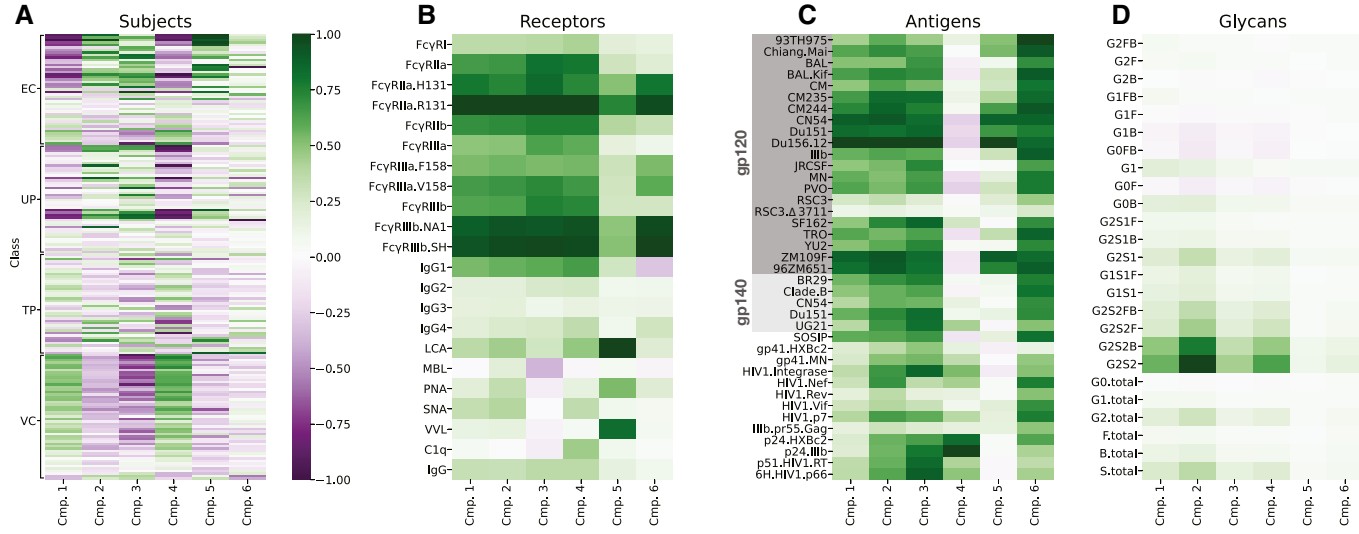

**Figure 5. Factor components represent consistent patterns in the HIV humoral immune response.**

A–D Decomposed components along subjects (A), receptors (B), antigens (C), and glycans (D). EC: elite controller, TP: treated progressor, UP: untreated progressor, VC: viremic controller (see Methods). All plots are shown on a common color scale after scaling each factor component within the range −1 to 1. Antigen names indicate both the protein (e.g., gp120, gp140, gp41, Nef, and Gag) and strain (e.g., Mai and BR29). Descriptions of each receptor name can be found in Table 1.

**Table 1. Descriptions of the receptor detections found within the tensor analysis**

| Receptor | Description |
|---|---|
| FcgRI | FcγRI (Brown *et al*, 2017) |
| FcgRIIa | FcγRIIa (Brown *et al*, 2017) |
| FcgRIIa.H131 | FcγRIIa.H131 (Brown *et al*, 2017) |
| FcgRIIa.R131 | FcγRIIa.R131 (Brown *et al*, 2017) |
| FcgRIIb | FcγRIIb (Brown *et al*, 2017) |
| FcgRIIIa | FcγRIIIa (Brown *et al*, 2017) |
| FcgRIIIa.F158 | FcγRIIIa.F158 (Brown *et al*, 2017) |
| FcgRIIIa.V158 | FcγRIIIa.V158 (Brown *et al*, 2017) |
| FcgRIIIb | FcγRIIIb (Brown *et al*, 2017) |
| FcgRIIIb.NA1 | FcγRIIIb.NA1 (Brown *et al*, 2017) |
| FcgRIIIb.SH | FcγRIIIb.SH (Brown *et al*, 2017) |
| IgG1 | Mouse anti-human IgG1 (Brown *et al*, 2012) |
| IgG2 | Mouse anti-human IgG2 (Brown *et al*, 2012) |
| IgG3 | Mouse anti-human IgG3 (Brown *et al*, 2012) |
| IgG4 | Mouse anti-human IgG4 (Brown *et al*, 2012) |
| LCA | Lens culinaris agglutinin (Brown *et al*, 2017) |
| MBL | Mannan binding lectin (Brown *et al*, 2017) |
| PNA | Peanut agglutinin (Brown *et al*, 2017) |
| SNA | Sambucus nigra lectin (Brown *et al*, 2017) |
| VVL | Vicia villosa lectin (Brown *et al*, 2017) |
| C1q | Human C1q (Brown *et al*, 2017) |
| IgG | Mouse anti-human pan-IgG (Brown *et al*, 2012) |

## Tensor method extensively reduces and visualizes dynamic responses to SARS-CoV-2 infection data

To demonstrate the general benefit of tensor methods in systems serology data analysis, we applied them to a separate dataset on acute SARS-CoV-2 infection (Data ref.: Zohar *et al*, 2020). In this dataset, samples from SARS-CoV-2-negative and -infected subjects were collected over the course of infection for about 4 weeks. Antibodies were tested for their antigen and Fc receptor engagement. We restructured the data into a three-mode tensor according to the sample, antigen, and receptor measured. In doing so, we obtained a tensor of size $438 \times 6 \times 11$ (Fig 6A). In this form, the tensor contains no missing values. After log-transforming and centering the data on a per-antigen–receptor basis, two components could explain 74% of the variance, with 0.3% the size of the original dataset ($[438 + 6 + 11] \times 2 = 910$ numbers; Fig 6B).

The resulting factors clearly separated into a clear acute (e.g., IgG3, IgM, and IgA) or long-term (IgG1-specific) response pattern (Fig 6C; Collins & Jackson, 2013), with the abundance of each program in each sample indicated by the sample factors (Fig 6E). Both components 1 and 2 generally shared broad specificity across antigens with slight differences (Fig 6D). As expected, component 1 also represented stronger association with FcγR and FcαR immune receptors (Fig 6C; Bruhns *et al*, 2009).

We proceeded similarly to earlier analysis (Zohar *et al*, 2020) and plotted each sample by the collection time after symptom onset, separated by the outcome of infection (SARS-CoV-2-negative, moderate disease, severe disease, and deceased; Fig 6F). A sigmoidal curve was fit to each temporal profile as a summary of the data. In contrast to the earlier analysis, we were able to plot these along the two components summarizing all the data, instead of the 66 individual measurements. Overall, samples showed a

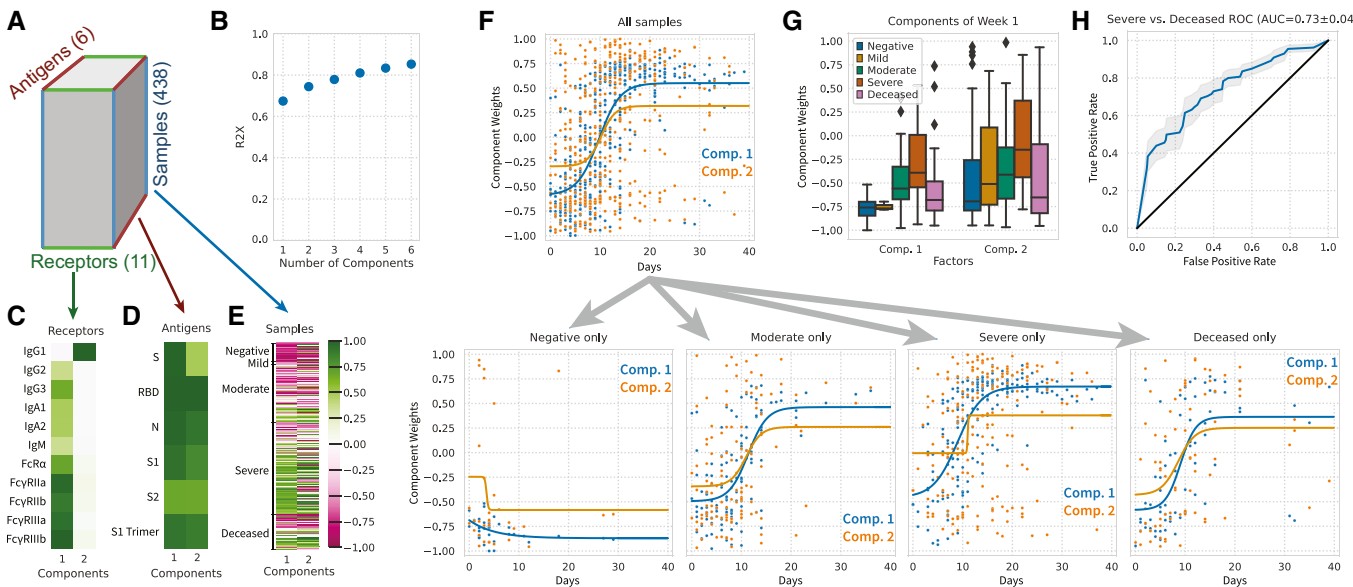

**Figure 6. Application of tensor factorization to SARS-CoV-2 systems serology measurements.**

A   Schematic of the data tensor. Measurements were arranged according to samples, target antigen, and receptor detection.

B   Percent variance reconstructed (R2X) versus the number of components.

C–E   Decomposed factor components along samples (C), antigens (D), and receptors (E).

F   Subject component weights plotted according to the sample time after symptom onset, together and separated into PCR-negative subjects along with moderate, severe, and deceased cases.

G   Boxplot of subject component weights for just samples within the first week after symptom onset, separated by subject group (negative $N = 33$, mild $N = 7$, moderate $N = 122$, severe $N = 196$, deceased $N = 74$). Each point represents a distinct biological sample. The three bands in each box represent the first, second, and third quartiles of the weights, from the bottom to the top, respectively; the whiskers extend up to 1.5 times the interquartile range beyond the box range; any outliers beyond the whisker ends are plotted as single points.

H   ROC curve of logistic regression classifier for predicting severe disease versus deceased outcome. Model is built using the two component weights of the subject factors. The shaded area indicates the standard deviation from a 10-fold cross-validation.

time-dependent increase in factor values (Fig 6F). Interestingly, a subset of PCR-negative subjects showed positive weights specific to component 2, indicating some IgG1-specific pre-existing immunity. As previously observed, severe cases displayed a component 1 response that, on average, had a higher initial and final quantity than either moderate or deceased cases (Fig 6F and G). A logistic regression classifier using just week 1 data predicted severe versus deceased outcome, with an AUC of 0.73 (Fig 6G and H), comparable with a random forest classifier in previous analysis (Zohar *et al*, 2020; AUC 0.71). Factorization with more components than 2 did not improve classification accuracy.

# Discussion

We show here that tensor-structured data decomposition can improve our view of systems serology measurements. Specifically, this approach recognizes that antibody variation takes place across the distinct and separable antigen binding and Fc/receptor dimensions. Using this property, we identify that these measurements can be reduced more efficiently (Fig 2), this reduction is robust to missing values (Fig 3), and properties of the immune system and infection can be accurately predicted (Fig 4). Most critically, this form of dimensionality reduction provides a clearer interpretation of the resulting models (Fig 5) as it accounts for the high degree of

intercorrelation across each dimension. Finally, reducing the data into patterns enables robust associations between the biophysical parameters of antibodies and functional responses or immunological status (Figs 4 and EV2).

The resulting factors and their association with infection state extend prior knowledge regarding changes in humoral immunity in HIV. One of the clearest patterns is an association of progression status with components 1 and 4, representing an antigen shift between surface and intracellular antigens (Figs 4F, 5C and EV1B). Abundance of p24 antigen and its antibody titer has been proposed as an effective marker of HIV progression (Schüpbach *et al*, 2000), and predictive of death (Rubio Caballero *et al*, 2000), although it correlates strongly with viral RNA and CD4+ counts (Sabin *et al*, 2001). As we observe with component 4, viremic controllers have been characterized as having especially high p24-specific IgG1 and IgG2 driving phagocytic responses (Tjiam *et al*, 2015). The negative association with component 1 likely reflects a decrease in antibody titers overall which has separately been found to predict progression (Tsoukas & Bernard, 1994). Therefore, although features of p24 abundance or antibody titers may have an incomplete and complex relationship with progression, a p24/gp120 ratio may be more predictive (Fig EV3). Viremia status was predicted through an even decrease in many of the components (Fig 4F), generally opposite the component weights predicting functional responses (Fig 4E). This broad difference matches perfectly with previous observations

that viral control is associated with polyfunctionality, rather than a specific molecular program (Ackerman *et al*, 2016). Predicting viremia using the viral RNA quantities, rather than classifying groups based on a threshold, could reveal more specific regulatory changes because, for example, elite controllers were heterogeneous as a group, and this variation may correlate with viral RNA amounts (Fig 5A; Côrtes, 2015).

The functional predictions and their component contributions matched expected patterns. All three NK activation measurements (CD107a, IFNγ, and MIP1β) had very similar weights to be expected, given their high correlation (Fig 4E). The only component with a negative weight with respect to gp120/gp140, component 4, showed a negative or negligible contribution to functional predictions (Figs 4E and 5C). More specifically, ADCC was predicted by positive association with all the components that included both FcγRIII and surface (gp120/gp140) antigen binding (components 1, 2, 3, and 6; Figs 4E and 5B and C). ADCD had only two consistently positive component weights—1 and 5—which can be taken to reflect probably overall titers and lectin pathway complement activation, respectively, as component 5 had unusually strong weights for the glycan-binding probes LCA, PNA, and VVL (Fig 5B) (Merle *et al*, 2015). In contrast, although sparse models can predict these functions accurately (Alter *et al*, 2018), one cannot assign significance to the individual model weights as they change upon resampling the dataset (Fig EV2), a challenge when modeling highly correlated measurements such as these (Candes & Tao, 2007; Efron, 2020; preprint: Tansey *et al*, 2021). Establishing links between specific molecular factors and functional responses could be further improved by experimentally introducing less correlated variation into the data, such as by measuring samples after enzymatic glycosylation modifications or depletion of certain isotypes (Albert *et al*, 2008; Chung *et al*, 2014). This also highlights the need for multivariate serological profiling as single-factor studies are likely to find indirect associations.

Although the glycan measurements had insufficient variation to link them to specific molecular programs beyond variation in amounts overall, future refinements to these measurements or analysis may reveal more precise regulation (Lofano *et al*, 2018). Glycan measurements across a panel of antigens might reveal more specific regulation, particularly as glycans are known to be tuned in an antigen-specific manner (Kaneko *et al*, 2006; Albert *et al*, 2008; Larsen *et al*, 2020). Paired glycan and biophysical measurements in acute infection may also reveal more drastic glycan variation, especially given links with outcomes such as between severe COVID-19 and IgG fucosylation (Larsen *et al*, 2020; Zohar *et al*, 2020). A tensor partial least squares regression approach would also reveal variation specifically associated with glycan changes by specifically focusing on variation shared in both datasets (Zhang & Li, 2017).

A more recent study examining SARS-CoV-2 infection allowed us to explore whether tensor-structured dimensionality reduction has benefits that extend to the serology of other disease and in longitudinal studies (Fig 6). Surprisingly, we found that just two patterns within these data could explain 74% of the variance (Fig 6B). Although we were able to replicate the difference in dynamics between severe and deceased cases (Fig 6F–H), the sufficiency of just two patterns argues for quantitative differences in these two patterns, rather than detailed qualitative changes in the immune response (Zohar *et al*, 2020). Perceived differences between

individual measurements could arise in part from these two component patterns being combined in each measurement. As evidence of this, the reported measurements that differed in dynamics between severe and deceased subjects were almost exclusively those we observed to be weighted on both components 1 and 2, but those specific to component 1 showed no difference between outcomes (Zohar *et al*, 2020; Fig 6E and F). It is also difficult to draw conclusions on a measurement-by-measurement basis, even in large studies such as this, due to large subject-to-subject variability and strong correlations between measurements (e.g., Fig EV2). On the other hand, it is possible that immunologically significant patterns remain in the unexplained variance that are drowned out by the most drastic changes (Larsen *et al*, 2020). There is also a challenge in separating pre-existing partial immunity from prior exposures or cross-reactivity with other coronaviruses; some PCR-negative cases showed positive weights on component 2, presumably indicative of long-term humoral responses, possibly from cross-reactivity with other coronaviruses (Fig 6F) (Ng *et al*, 2020). Longer term longitudinal studies of acute infection would allow one to observe the transition from acute immune response to lasting protection and potentially better resolve the dynamics of class switching alongside its functional consequences (Collins & Jackson, 2013).

Other tensor arrangements of serology data will help reveal new patterns within these data. Indeed, here, we have arranged data both with subject, antigen, and receptor modes, in either a coupled form (Fig 1) or a single tensor (Fig 6A). With longitudinal data in which time points can be aligned, one could create a mode representing the contribution of time (Chitforoushzadeh *et al*, 2016; Farhat *et al*, 2021). Although each antigen is treated similarly along one dimension, antigenic mutants or strains could also be separated into separate tensor modes before decomposition. This could lead to further data reduction (e.g., both strains of p24 and gp41 antigens share a similar signature; Fig 5C) and simplify comparisons between strains. Differences in the weights would also essentially serve as an unsupervised prediction for competition experiments to reveal differences in the binding targets of polyclonal serum (Georgiev *et al*, 2013). As compared with traditional blocking or mutational experiments, antigens in these measurements are multiplexed across identifiable beads (Angeletti *et al*, 2017; Sesterhenn *et al*, 2019). Therefore, making measurements across a wider antigen panel requires just small amounts of each antigen and can be scaled to hundreds of antigens without increased sample requirements. Finally, CMTF could be used to link other types of immune response measurements besides glycan quantitation to serology, such as cytokines and gene expression.

More effective dimensionality reduction in turn enables new ways of viewing antibody-mediated protection. Thinking of these measurements as akin to the microarray for gene expression data suggests new possibilities in leveraging these data. One valuable property of CMTF is that it separates the immune receptor and antigen-binding patterns within the data. This will enable surveys for common Fc response patterns across diseases and studies because these different datasets would still share this axis. This "transfer learning" could therefore help identify common patterns of immune dysregulation. With more extensive profiling of the various glycosylation and isotype Fc forms, it would be possible to fix the receptor axis of the decomposition, in effect matching new measurements to specific known immunologic patterns. These

pattern-matching approaches would be much like gene set enrichment analysis for expression data (Subramanian *et al*, 2005). The binding interactions of antibodies, though producing combinatorial complexity, are a simple set of antigen and receptor binding. Ultimately, one should be able to apply multivalent binding models to mechanistically model the interactions within serum (Perelson & DeLisi, 1980; Robinett *et al*, 2018; preprint: Tan & Meyer, 2021). This might allow separation of avidity versus affinity in binding and integration with extensive prior characterization of Fc properties, such as the biophysical properties of individual glycoforms and isotypes (Bruhns *et al*, 2009; Dekkers *et al*, 2017). A mechanistic view could also help guide more advanced multimodality therapeutic interventions, like inhibitors or enhancers of antibody response that cooperate with the cocktail of endogenous antibodies (Kaneko *et al*, 2006; Pagan *et al*, 2018).

Ultimately, a comprehensive view of immunity needs advancements in measurements that are complementary to systems serology. Much like how systems serology has served to profile antibody-mediated protection, profiling methods are helping to characterize T-cell-mediated immunity (Birnbaum *et al*, 2014). These technologies, alongside more traditional technologies to profile cytokine response, gene expression, and other molecular features, promise to provide a truly comprehensive view of immunity. Integrating these data will require dimensionality reduction techniques that recognize the structure of these data alone and in combination. Factorization methods, especially those operating on tensor structures, will be a natural solution to this challenge, due to their scalability, flexibility, and amenability to interpretation (Omberg *et al*, 2007).

# Materials and Methods

## Reagents and Tools table

| Reagent/resource | Reference or source | Identifier or catalog number |
|---|---|---|
| **Software** | | |
| **TensorLy Python Library** | http://tensorly.org/ | v0.6.0 |
| **SciPy Python Library** | https://www.scipy.org/ | v1.7.0 |
| **NumPy Python Library** | https://numpy.org/ | v1.21.0 |
| **Pandas Python Library** | https://pandas.pydata.org/ | v0.2.5 |
| **Seaborn Python Library** | https://seaborn.pydata.org/ | v0.11.1 |
| **Python** | https://www.python.org | v3.9.5 |

## Methods and Protocols

### Subject cohort, antibody purification, effector function assays, and glycan analysis

All experimental measurements were collected from prior work (Alter *et al*, 2018; Data ref.: Zohar *et al*, 2020). Measurements were clipped to be at least 0.1 (HIV glycan), 1.0 (HIV biophysical), or 10.0 (SARS-CoV-2 biophysical); log-transformed; and then centered on a per-measurement basis across subjects. The thresholds before log-transformation were determined to be well below the level of noise in the assays using the negative controls for each. Two antigens, gp140.HXBc2 and HIV1.Gag, were identified to only have one and two receptor measurements, respectively, making their factor values unstable because almost all measurements were missing. These were removed on import during the tensor-based analysis. HIV subjects were classified into four categories: untreated progressors, who failed to control viremia without combined antiretroviral therapy (cART); treated progressors, who similarly failed to control viremia without cART but were on it for the study measurements; viremic controllers, who possessed a viral load between 50 and 2,000 RNA copies/ml without cART; and elite controllers, who had < 50 copies/ml without cART. These were then grouped into two classifications: controllers (EC and VC) versus progressors (UP and TP); and viremic (UP and VC) versus nonviremic (TP and EC).

### Coupled matrix–tensor factorization

We decomposed the systems serology measurements into a reduced series of Kruskal-formatted factors. Tensor operations were performed using Tensorly (Kossaifi *et al*, 2019). Most measurements were made across specific antigens, and we structured them into a three-mode tensor, $\mathcal{X}$, whose modes represent subjects, receptors, and antigens. Separately, gp120-associated antibody glycosylation was measured for half of the HIV subjects. These measurements were structured into a matrix, $Y$, representing the quantities for each subject (Fig 1).

Shaping antigen-specific data into a three-mode tensor recognizes that measurements of the same receptor or antigen should share variation within each component. However, as not all receptor–antigen pairs were measured, the constructed tensor contained missing values from the perspective of this data structure. Throughout the factorization algorithm, we used censored least squares solving, with rows corresponding to missing values removed.

In preprocessing, we scaled the matrix so that it contained the same amount of variance as the tensor. To perform CMTF, we assumed the subject mode was shared between the tensor and the matrix:

$$\mathcal{X} \approx \sum_{r=1}^{R} \mathbf{a_r} \circ \mathbf{b_r} \circ \mathbf{c_r} = \widehat{\mathcal{X}}$$

$$Y \approx \sum_{r=1}^{R} \mathbf{a_r} \circ \mathbf{d_r} = \hat{Y}.$$

Here, "∘" represents the vector outer product and $R$ is the total number of components in the factorization. The original tensor is approximated as a sum of $R$ rank-one tensors constructed by the vector outer product along each mode. The original matrix is represented by the sum of $R$ rank-one matrices formed by the outer product of row and column vectors. For the $r$-th component, $\mathbf{a_r}$, $\mathbf{b_r}$, and $\mathbf{c_r}$ are vectors indicating variation along the subject, receptor, and antigen dimensions, respectively, and $\mathbf{d_r}$ is a vector indicating variation along glycan forms within the glycan matrix.

Decomposition was initialized using singular value decomposition of the unfolded data along each mode, with missing values imputed by a one-component PCA model and entirely missing columns removed. We then optimized the decomposition using an alternating least squares (ALS) scheme (Kolda & Bader, 2009) for up to 2,000 iterations. In each ALS iteration, linear least squares solving was performed on each mode separately (preprint: Acar *et al*, 2011; Battaglino *et al*, 2018):

$$\min_A \| [X_{(1)}Y] - A[(C \odot B)^T D^T] \|^2$$
$$\min_B \| X_{(2)} - B[(C \odot A)^T] \|^2$$
$$\min_C \| X_{(3)} - C[(B \odot A)^T] \|^2$$
$$\min_D \| Y - AD^T \|^2$$

where $X_{(1)}$, $X_{(2)}$, and $X_{(3)}$ are the tensor unfoldings of $\mathcal{X}$ along each mode, and "⊙" represents a Khatri–Rao product. The R2X was checked on each even iteration, and decomposition was terminated early if the change was found to be $< 10^{-5}$.

### Justification of multiplicative factor interactions

Kruskal-formatted tensors are structured such that each factor component (receptors, antigens, subjects) should be multiplied together to reconstruct the data. This structure is simply a higher dimensional generalization of matrix decomposition techniques like PCA or non-negative matrix factorization, in which scores and loadings matrices are multiplied together to reconstruct the data. An expectation of these approximations is that variation within the tensor occurs in a pattern that can be localized to each tensor slice, which is justified by the nature of the measurements being considered. These measurements are made in an array format, wherein plasma samples from subjects are incubated with individually identifiable beads covalently conjugated with antigens (Brown *et al*, 2012, 2017). The conjugated antigens isolate IgG fractions specific to those targets. After washing, the beads are incubated with fluorescently labelled detection reagents that bind to the isolated IgG depending upon their properties. Thus, in essence, the assay is a bead-based sandwich ELISA, in which IgG is the sandwiched target. Given the format, the amount of fluorescence measured on a given bead should be proportional to both (i) the amount of IgG isolated on the bead and (ii) the fluorescence signal obtained per isolated IgG (two of the tensor modes), supporting their multiplication. A multiplicative relationship allows each factor to contribute positively, negatively, or not at all to the variation represented by a component by a positive, negative, or zero weighting. A strong

validation of this structure in a tensor form is the large extent to which the data can be reduced without loss of information. It also fits with the biological expectation that antibody Fab variation influences the data along the antigenic slices, whereas Fc variation influences the data along the receptor ones.

### Reconstruction fidelity

To calculate the fidelity of our factorization results, we calculated the percent variance explained, R2X. First, the total variance was calculated by summing the variance in both the antigen-specific tensor and the glycan matrix, where included:

$$v_{total} = \| \mathcal{X} \|^2 + \| Y \|^2.$$

Variance was defined as the sum of each element squared, or the square of the norm. Any missing values were ignored in the variance calculation throughout. Then, the remaining variance after taking the difference between the original data and their reconstruction was calculated:

$$v_{r,antigen} = \| \mathcal{X} - \widehat{\mathcal{X}} \|^2$$

$$v_{r,glycosylation} = \| Y - \hat{Y} \|^2.$$

Finally, the fraction of variance explained was calculated:

$$R2X = 1 - \frac{v_{r,antigen} + v_{r,glycosylation}}{v_{total}}$$

where indicated as Q2X instead, this quantity was calculated only for values left out to assess the fidelity of imputation.

### Logistic regression/elastic net

The data were centered and variance-normalized prior to model assembly. Logistic regression and elastic net were performed using `LogisticRegressionCV` and `ElasticNetCV` implemented within scikit-learn (Pedregosa *et al*, 2011). Both methods used 10-fold cross-validation to select the regularization strength with smallest cross-validation error and a fraction of L1 regularization equal to 0.8 to match previous results (Alter *et al*, 2018). Logistic regression used the SAGA solver (preprint: Defazio *et al*, 2014).

### Cross-validation

We employed a 10-fold cross-validation strategy to evaluate each prediction model. Subjects were randomly assigned to folds to prevent the influence of subject ordering in the dataset. We found that sharing the cross-validation fold structure between hyperparameter selection and model benchmarking led to consistent overfitting. Therefore, we used a nested scheme in which the folds were assigned differently for hyperparameter selection and model performance quantification.

### Principal components analysis

Principal components analysis was performed using the implementation within the Python package `statsmodels` and the SVD algorithm. Missing values were handled by an expectation–maximization approach wherein they were filled in repeatedly by PCA. This filling step was performed up to 100 iterations or until convergence as determined by a tolerance of $1 \times 10^{-5}$.

### Missingness imputation

To evaluate the ability of factorization to impute missing data, we introduced new missing values by removing (i) entire receptor–antigen pairs or (ii) individual values from the antigen-specific tensor as indicated and then quantifying the variance explained on reconstruction (Q2X). More specifically, in the first situation, fifteen randomly selected receptor–antigen pairs were entirely removed (2,715 values) and marked as missing across all subjects, leaving ~ 93,000 values for training. In the second, fifteen randomly selected individual values were removed, leaving ~ 96,000 training values. CMTF decomposition was performed in each trial as described before, and the left-out data were compared with the reconstructed values. There were 20 or 10 trials performed in each imputation situation, respectively. Varying numbers of components were used for decomposition, and a Q2X was calculated for each. In the second case, we compared CMTF with PCA-based imputation with the dataset flattened into a matrix form.

### Fitting sigmoidal curves

The sigmoidal curves in Fig 6F were fit to $y = A/[1 + \exp(-k(x - x_0))] + C$ using the Levenberg–Marquardt algorithm as implemented within `curve_fit` in the Python package `scipy`. The initial optimization point was set so that $A$ was 0.6 of the $y$ range; $C$ was the smallest $y$; $x_0$ was the median $x$; and $k$ was either 0.5 or −0.5, depending on whether the mean of the first half of $y$'s was larger or smaller than that of the latter half.

## Data availability

The computer code produced in this study is available on GitHub (https://doi.org/10.5281/zenodo.5184449).

**Expanded View** for this article is available online.

## Acknowledgements

This work was supported by NIH U01-AI-148119 to A.S.M.

## Author contributions

ASM conceived of the study and wrote the original draft. ASM and ZCT curated the data and developed the method. ASM, ZCT, MCM, and HSA performed the computational analysis. ZCT, MCM, HSA, SDT, and ASM visualized the outcomes, discussed the results, and edited the paper.

## Conflict of interest

The authors declare that they have no conflict of interest.

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
