## [Review Process File · Molecular Systems Biology]

Tensor-structured decomposition improves systems serology analysis

Zhixin Tan, Madeleine Murphy, Hakan Alpay, Scott Taylor, and Aaron Meyer

DOI: [10.15252/msb.202110243](https://doi.org/10.15252/msb.202110243)

Corresponding author(s): Aaron Meyer (a@asmlab.org)

Review Timeline:

Submission Date:	22nd Jan 21
Editorial Decision:	17th Mar 21
Revision Received:	20th Jul 21
Editorial Decision:	10th Aug 21
Revision Received:	12th Aug 21
Accepted:	16th Aug 21

Editor: Maria Polychronidou

Transaction Report:

RE: MSB-2021-10243, Structured decomposition improves systems serology prediction and interpretation

Thank you again for submitting your work to Molecular Systems Biology. We have now heard back from the three referees who agreed to evaluate your study. Overall, the reviewers acknowledge that the presented approach seems potentially relevant. They raise however several substantial concerns, which preclude the publication of your study in its current form.

After consulting with the reviewers during our referee cross-commenting process (in which the reviewers can make additional comments on each other's reports), we have decided to offer you a chance to address the issues raised in a major revision.

Without repeating all the points listed below, some of the more fundamental issues are the following:

- Reviewers #1 and #3 point out that the proposed benefits of the approach i.e. improved prediction performance and improved interpretability need to be more convincingly supported by further analyses and statistical tests. All reviewers provide constructive suggestions in this regard and indicate specific cases where claims are not well supported by the data.
- The reviewers raise several technical concerns (e.g. regarding centering the data, the structure of the data, imputing missing values etc.). These concerns need to be carefully addressed as they potentially undermine the relevance and significance of the main conclusions.

All issues raised by the reviewers need to be satisfactorily addressed, and the proposed methodological advance needs to be well supported by the data and analyses. As you may already know, our editorial policy in principle allows a single round of major revision. It is therefore essential to provide responses to the reviewers' comments that are as complete as possible.

Reviewer #1:

Summary

Murphy et al report results of an alternative dimensionality reduction approach (involving tensors) to consolidate information in systems serology data sets (biophysical antibody profiles) in predicting antibody activities and subject class information. They build on a published data set and more straightforward approach previously reported in MSB.

The principle claims are that total tensor-matrix factorization "outperforms standard methods like PCA in the extent of reduction possible", and that reduction "improves prediction" and improves interpretability.

General remarks

Unfortunately, evidence to support one of these claims is not provided (improved prediction performance), and another is subjective (interpretability). Thus, while it is exciting to see this approach (considered a bit of step toward a deep learning approach) employed on this data, the manuscript requires substantial revision in terms of these claims. The methods swap is not proven to be an advance. It is more accurate to describe it as an equivalent but promising substitution. The work's contribution is purely technical, shows similar performance, and some would find interpretation is more rather than less challenging.

Major points

1. I think it is a matter of opinion that per-measurement labels make for more complicated interpretations and that tensors are fundamentally more interpretable. Language claiming this should be considerably softened and focused more on what the distinct value is to this approach than on its superiority (which is subjective).

2. Good examples of meaningful interpretation of TMTF components are given, but the expertise required to do so is not trivial, and the scaling choices complicate matters. It is important to note that these examples represent "just so" stories apparent in the original approach, and that there

are also examples that have no known biological basis, as well as those that contradict expectation from prior knowledge that can be found in the data.

!!!!3. The authors claim results are "significantly" improved. Fig 5 shows no statistical tests to support this claim, and data that appears to best show equivalence. This claim needs to be proven or eliminated. (A prior version of the manuscript available on bioarxiv reports what might be considered to be fundamentally different results in this figure. They appear to have updated their observations but not the language describing them.)!!!

4. The distinction between prior approaches and this approach is overstated in some ways. While the authors state that "optimal models required 30-50 variables" in the prior approach, similar performance results from considerably sparser models. There are, however, still fewer features used here, in exchange for some considerable barriers to interpretation.

5. Fv, Fc, and antibody function dimensions have been collapsed intentionally by others (Fv and Fc breadth and polyfunctionality scores, for example). These alternative approaches, which marginally reduce novelty should be mentioned.

6. Why not just go all the way to a deep learning approach?

Minor points

1. Antibody glycan data in this study included antigen-specific antibody glycan profiles, not "antigen-generic" glycan data. The text repeatedly describes this underlying data incorrectly.

2. The introduction states: "To date, no methods have provided a means to holistically visualize the variation in serology measurements." This claim is overly broad and so vague as to be meaningless. It is not clear how standard global visualizations (such as a heatmap) do not meet this standard, and how the method proposed here (which is a dimensionality reduction approach) does accomplish this goal. The claim should be eliminated or properly bounded/specified.

3. What the authors consider to be "missing" data should be clarified. By their definition, this data set might be considered to have infinite missing data, as for example, no T cell data was reported. It feels misleading to characterize measurements never assessed as missing. The information they provide is confusing to someone very familiar with this data set, and likely not meaningful to individuals unfamiliar to the data set. Perhaps either greater or lesser detail in the opening section would be an improvement to the current level (which is clarified subsequently).

4. Why does Fig 2a not report variance reconstructed with an alternative method to TMTF? What effect does the normalization have on the comparison between PCA and TMTF in Fig 2b? Is the normalization to the same value for both PCA and TMTF, or does it differ between them?

5. Figure 3 aims to show accuracy, but reports variance predicted and provides no comparison to alternative approaches.

6. The authors claim that patterns in env and internal antigen binding would be "nearly impossible" to identify without their approach. Antigen-specificity differences are typically readily apparent in humoral response data sets. The authors should prove this assertion or eliminate/soften it.

7. Why are randomization results not shown for Fig 5a? (or, eek, are they sitting underneath other points?)

8. The supplemental data provides little value.

Reviewer #2:

Summary

The manuscript describes a new form of tensor factorization, total tensor-matrix factorization (TMTF), with application to systems serology measurements to improve the prediction of immune functional responses, classification of subjects according to their HIV control status and

interpretations of models and predictions. TMTF capitalizes on the ability to separate the contribution of antigen from immune receptor binding that ultimately increase data reduction by avoiding repetition of the antigens for each receptor measurement. The approach is evaluated with an example systems serology study, previously published by Alter, G. et al which profiled immune responses to predict both functional immune responses and disease status within HIV-infected subjects. The authors show that TMTF reduced the dimension (to 2.8%) of the data while preserving about 93% of the original variation. The accuracy of TMTF decomposition in predicting functional measurements and subject viral and controller status is shown to be comparable to results published in Alter, G. et al. The authors present an interesting concept to expand on systems serology methods which handle highly dimensional and correlated data. Since the study explains a new approach, the manuscript should refocus the writing to describe specifics of this approach in order to support the benefits highlighted by the authors.

Major comments:

1. The authors should describe details of the approach for generalized systems serology data. Details on data requirements (i.e. preparation or pre-processing) should be included. Parameter tuning; how is the optimal number of components selected? How is the method evaluated in terms of performance and stability? How is the quality of imputation evaluated?
2. The referenced paper for the data (Alter, G. et al) differentiated the subjects across the four groups with approximately 60% accuracy in addition to the prediction of progression and viremia classes. It would be useful to evaluate how well TMTF classifies the four groups. It is surprising that Fig. 4a does not show separation of classes in any of the component although the text suggests an improvement in the subject class prediction.
3. What was the rationale for not standardizing the measurements? TMTF, like PCA, is a variance maximizing approach, it is expected that measurements with highest variance will dominate the components. This was recognized by the authors for measurements of gp140 HXBc2 (mislabelled as HVBc2) and was corrected by multiplying by 0.000001 to scale down this measurement. Not standardizing the data may have contributed to the inability to identify IgG subclass patterns as a result of FcγRs responses overwhelming IgG subclass responses. In Alter, G. et al, IgG subclass assessment revealed that IgG3 contributed to ADCD activity while IgG2 and IgG4 impeded its activity but TMTF did not identify any of the patterns associated with IgG subclass.
4. Is the order of components important? Given that Component 1 describes greater than 75% of the variance, it is surprising that the component is not mentioned anywhere in the text and did not correspond to an important immune pattern. How are the components related? For example, are they orthogonal like in PCA? The description of the data seems to be treating the components entirely as stand alone structures.
5. Prediction of ADCD was exclusively explained by Component 5 which is interpreted by the authors to be indicative of the lectin-pathway complement activation, shown by positive weighting of LCA, PNA and SNA responses. In contrary, in Fig. 4b, it appears that Component 5 is dominated by mostly FcγRs than LCA, PNA and SNA responses.

Minor comments:

1. The writing is overly generalized with background material on systems serology repeated across different sections of the manuscript. It will be useful to reduce background in the abstract (which is currently half of the abstract) and expand on the key aspects of the approach that contributes to systems serology. Background material is again repeated on the results section as the entire first paragraph of "Factor components represent consistent patterns in the HIV humoral immune response". The results section needs to be rewritten as a large fraction of the text correspond to interpretation or discussion points, this needs to describe the results presented in the figures.
2. The authors should describe the importance of the patterns highlighted in the results with

- respect to HIV immunity? For example, Gag-binding IgG1 in Component 4 and the shift between surface antigen (gp120, gp140) and p24/p51/p66 binding in Component 5.
3. Fig. 4b shows that the components are dominated by the entire range of FcyR responses (apart from Component 4), so it is not very clear why immune functions are predicted by some components and not others.
 4. The authors should support results with data and avoid using the word 'believe'. Component 2 is described to represent broader antigen binding which 'explains the positive correlation with all glycoforms.' These positive correlations are not clearly shown by data in Fig. 4d.
 5. Imputation was tested by removing entire receptor-antigen pairs across all subjects. Could TMTF impute measurements missing in only a few subjects? How much missing data could TMTF handle while maintaining accuracy?
 6. How is imputation evaluated? It seems strange that imputation (Fig. 3) seems to perform much better than original complete data (Fig. 2a)
 7. Remove subscript "a" in Fig. 3
 8. It would be interesting to compare to partial least squares (PLS) which also maximizes the covariance between components from two data sets to extract an underlying structure.
 9. What does feature importance quantify in Fig. 5? It is hard to interpret feature importance for class prediction. Does it mean that Component 1 could accurately predict 75% progressors and component 5 could predict 100% of progressors and 100% viremic subjects? Why is there no discrimination of progressors in Component 5 of Fig. 4a.
 10. CMTF in Fig. 1 is not previously defined
 11. Fig. S1 is not referenced anywhere in the text
 12. The word 'Component' is sometimes capitalized and sometimes not capitalized.

Reviewer #3:

This manuscript develops a form of tensor factorization that incorporates both direct immune interactions (e.g. antibody-virus binding) as well as secondary modes of immune defense mediated by effector cells (e.g. natural killer cells, complement). One key advantage of this method is its ability to incorporate disparate datasets, such as subject-antigen-receptor and subject-glycosylation data, while retaining the underlying structure of each data set. The authors apply this framework to an HIV systems serology study and demonstrate that this vast data set is low-dimensional, permitting ~90% of the variance to be reconstructed using 10 components.

While I found the core concept in this paper very interesting, I was confused about one critical issue: whether the data was centered prior to analysis. If not, then the ~90% variance explained could be a description of how far away the data lies from zero rather than how well the model characterizes the data. As stated below, the authors should clarify this point (both for the TMTF and PCA analysis). In addition, there are a few areas of improvement that would help clarify the main results of this manuscript and quantify how robust their results are.

Major comments:

1. Centering the data: Our understanding is that the data used in the factorization algorithm is not centered, and therefore the variance explained by the first factor describes how uncentered the data is instead of how well the model characterizes the data [for a visual example, see <https://i.stack.imgur.com/9vENg.jpg>]. This could artificially inflate their variance explained in Figure 2A and Figure 3, and the authors should rerun their analysis on centered data to see how it affects their analysis.

1.1 There should be similar clarification for the PCA algorithm used as a point of comparison. Was the data centered before PCA (some python functions automatically do this)?

1.2 Beyond the issue of centering, the authors should clarify why the PCA results were different from the TMTF results. Was it caused by the fact that in TMTF the eigenvectors do not need to be orthogonal, or did TMTF retain the structure of the antigen-receptor pairs while the PCA analysis lost this structure when the tensor was unfolded?

2. Implied structure of the data: The results of the analysis are presented nicely in Figure 4. However, it is unclear what inherent assumptions were made when forcing the data to conform to a simple multiplicative model.

2.1 For example, if the subject-receptor-antigen data were converted into different units (e.g. from Molar to $\mu\text{g/mL}$ units), the framework could perfectly adjust by multiplying all receptor (or antigen) coefficients by the conversion factor. However, if the subject-receptor-antigen data was inverted while not inverting the glycan data (e.g. if binding was reported using association constants K_A rather than dissociation constants $K_D=1/K_A$), then a different set of receptor and antigen values would be inferred. These values could characterize the data much better or much worse, but the authors chose one specific representation for their analysis. Is there some structure to the data that suggests this particular setup?

2.2 An additional question is whether the receptor and antigen values shown in Figure 4 represent true biological properties (i.e. whether these inferred values would be the same for other data sets), or if they are dataset-specific. For example, if you either: (1) restrict the analysis to different subsets of the data (e.g. 50% of the receptors and 50% of the subjects/receptors/antigens) or (2) remove the glycan data entirely, do you recover similar antigen and receptor values? Ideally, this question would be tested against a separate data set with overlapping antigen-receptor pairs (if one is readily available), but analyzing different subsets of the current data set will suffice.

2.3 "This component also displays negative weighting across gp120 and gp140 antigens, likely reflecting a decrease in antibody titers overall." This is an important point that is worth elaborating upon. How would a difference in antibody titer show up in the data? Wouldn't it decrease the Subject factor in Figure 4A, since you would expect all measurements to be lower for that subject?

2.4 The authors also allude several times to potentially non-linear interactions in the immune system, such as when multiple antibodies interact with one another. Indeed, when analyzing the immune functionalities in Figure 5, the authors use a Gaussian process model to account for non-linearities. Aside from Figure S2, are there other indications of non-linearity in the data? Conversely, are there other pairs of measurements in Figure 4 (analogous to Figure S2) that confirm the linearity assumption?

2.5 Is there a hyperparameter to quantify the relative importance of the antigen-receptor vs glycan data? If so, how is its value determined? For example, components #1 and #2 look nearly identical for the glycans in Figure 4D, suggesting that these two components primarily differentiate the antigen-receptor data.

3. Figure 3:

3.1 I was surprised that the variance explained by a single component is so high, especially given the lower (and more slowly growing) R^2 in Figure 2A. Is this related to the centering issue, or was it because so few measurements were left out in the Q2X analysis? The authors should clearly state the number of entries (or percentage of entries) that were available for training and the number that were imputed.

3.2 The authors chose 15 receptor-antigen pairs to entirely remove and then reconstruct, but why not run this analysis across all possible receptor-antigen pairs to ensure that those 15 were not special cases? It would also be interesting to remove multiple pairs and see how the reconstruction error changes as the number/percent of missing measurements grows.

3.3 If this analysis is run multiple times on different subsets of left-out data, it would help to draw bands of uncertainty around each measurement (which may even out the dip in Figure 3 at 8 components). It might also be interesting to expand Figure 3 by drawing multiple curves showing different fractions of left out measurements (e.g. 1%, 5%, 10%, 50%...).

3.4 Could you put the Q2X value in perspective relative to experimental error? For example, when using 1 component, how far off (in absolute units) is the average imputed measurement from the actual value, and how does this correspond to typical measurement error?

4. Figure 5 and 6:

4.1 I am confused by the approach taken towards these functional measurements. The 10 components were derived from the Figure 4 data, so why should combinations of these components explain these additional data of ADCC, ADNP...? Perhaps a different set of components would be better suited to understand these functional measurements using a simple multiplicative model without resorting to a Gaussian process?

4.2 Were the "Alter et al" and "Randomized" methods in Figure 5 ever discussed? Was the "Randomized" method even used in Panel A (and the plot markers are hidden behind), or does that method only apply to Panel B?

=====

Minor comments: [These are all of the take-it-or-leave-it variety]

1. Figure 2B: It was difficult to parse what the "size of factorization" means or interpret why it is written in powers of 2.

1.1 It would help to state that the size of factorization in TMTF equals $(181 + 22 + 41 + 25)$ (# of components) and to give a similar statement for PCA.

1.2 Writing the x-axis as powers of 2 seemed arbitrary, and when I first saw this figure, I interpreted the 210 factorization as corresponding to TMTF with 10 components.

2. Results, First Section:

2.1 When first introducing matrix-tensor factorization, it would help to explicitly state that you are assuming a multiplicative model where each subject, antigen, receptor, and glycan are given some value, and the measurement for a specific subject-antigen-receptor or subject-glycan entry is given by the product of their factors. The first two equations in the Methods do this very well, so consider either bringing those into the main text or describing those equations and pointing the reader to the Methods.

2.2 Even more importantly, you should explicitly describe what the 10 different components represent, since they are so critical to the paper's message. The first time I read the manuscript, I did not understand (and hence ignored) the visual representation in Figure 1C, and the explanation only truly landed when I saw the equations in the Methods.

2.3 Regarding the first two equations in the Methods, explicitly define r as summing over the different components and R as the total number of components (presumably $R=10$ for Figure 4).

3. Figure 4:

3.1 Is there an intuitive explanation for why Components 3, 8, and 9 in Panel A are all uniformly purple? Is there a reason why Components 1, 2, 5, 7, and 10 are uniformly green? There is a lot of information in these figures, and it would help to pick out these prominent features that are immediately seen and explain them.

3.2 How is the IgG row in Panel B different from the IgG1, IgG2, IgG3, and IgG4 rows?

3.3 Do the rows that are nearly completely white (such as the rows from IgG2 to C1q in Panel B) arise because the data for all antigens and subjects against these receptors is zero? For example, I would have expected some signal from the other IgG subclasses (although IgG1 is expected to be

the strongest).

3.4 It would also help to know what the values mean for the antigen-receptor and glycosylation data. Is 0.5 a large value, and if so, what does it represent? Are there units associated with these values, or are they dimensionless fluorescence measurements?

4. Results, Fourth Section:

4.1 "and level of natural killer cell activation determined by expression of" → "and the level of natural killer cell activation represented by the expression of". It would also help to state that these are cumulative properties of the entire immune response to distinguish them from the individual factors in Figure 4.

4.2 "Separately, we predicted broad subject disease statuses: Controller versus Progressor, and Viremic vs Non-Viremic..." It felt jarring to read about this separate task before hearing about the functional data analysis (discussed in the previous sentence). Also, it would help to explicitly state how this classification is done. Do you learn which components are important by using a subset of subjects and then inferring the classifications for the left-out subjects?

4.3 "We therefore were able integrate the glycan data" → "We therefore were able to integrate the glycan data"

5. Figures 5 and 6:

5.1 It would be clearer if the labels in Panel B were "Viremic/Non-Viremic" and "Controller/Progressor" rather than only listing one.

5.2 As a minor point, it would be nice if the plot colors and plot markers were more consistent. For example, TMTF could be represented by black dots in Figures 2, 3, and 5. It is a small point, but it helps the reader associate the information between plots.

6. Results, Sixth Section: "Given the importance of component 6 in predicting nearly all of the functional measurements (Fig. 6A), its importance in predicting viremia is to be expected." This is an odd statement, given that Component 6 was not important for the Progression attribute [whereas according to this sentence, it should have been].

7. Discussion:

7.1 "just 8-10 consistent patterns..." Why do you give this particular range of values?

7.2 "While each functional measurement was predicted through a combination of factors, component 6 contributed to nearly every prediction (Fig. 4B). This suggests that these functions are tuned through both shared and individualized regulatory changes." Is this reference supposed to be for Figure 6B? I do not understand why you are specifically calling out component 6 but no other components. I also don't understand what the second sentence is stating.

7.3 "While each antigen is treated similarly along one dimension, antigenic sites and strains could be separated into distinct dimensions before decomposition." This is a very interesting point, and it would be worth elaborating upon. Is this possible to do with current measurements, or would it require additional experiments (and if so, how would these experiments separate out the binding contributions to each site)? Both blocking experiments [Sesterhenn 2019, <https://doi.org/10.1371/journal.pbio.3000164>] or mutational experiments [e.g. Angeletti 2017, <https://doi.org/10.1038/nr3680>], would be very time- and resource-intensive.

7.4 "Finally, the binding interactions of antibodies, while they produce combinatorial complexity, are a simple set of antigen and receptor binding." As with the previous comment, this is very intriguing. Would you need separate measurements for each monoclonal antibody, or could you decompose measurements from polyclonal serum [similar to Georgiev 2013, <https://doi.org/10.1126/science.1233989>]?

8. Reference 12: The title of this paper is missing "HIV" at the end. The other references should similarly be checked.

Point-by-point response to reviewers

First, we would like to thank the reviewers for their detailed and insightful comments. We hope this reflects the enthusiasm of the reviewers for our approach. We have endeavored to address each major and minor comment below through modifications of the manuscript and are confident that you will find it substantially strengthened by this effort. Many of the concerns touch on complex issues; we apologize for the length of this correspondence in advance. We look forward to your evaluation of these changes and explanations.

We wish to highlight a few overarching changes in particular:

- Prediction and imputation performances are now evaluated over multiple trials to eliminate the contribution of run-to-run variation.
- We have streamlined the text, with careful attention to placing background material, observations from our analysis, and resulting discussion in the appropriate sections.
- In both the manuscript and this response, we explain our normalization choices and some of the ways in which tensor-formatted analysis can differ from two-dimensional approaches like principal components analysis.
- We have added extended analysis to show benefits in interpretability, including two new Extended Data figures, and avoid just describing our conclusions. We have updated our language to reflect that our prediction accuracy is equivalent.
- To drive home the general utility of our approach, we have added an independent serology dataset from COVID-19 subjects, showing the same benefits as demonstrated in our initial HIV serology analysis.
- We identified that a pre-existing implementation we had used for CMTF from the Python package Tensorly was flawed, and the data reduction we implemented is in fact equivalent to a correct implementation of CMTF. We have therefore updated our language throughout the paper to reflect this.

Reviewer #1:

Summary

Murphy et al report results of an alternative dimensionality reduction approach (involving tensors) to consolidate information in systems serology data sets (biophysical antibody profiles) in predicting antibody activities and subject class information. They build on a published data set and more straightforward approach previously reported in MSB.

The principal claims are that total tensor-matrix factorization "outperforms standard methods like PCA in the extent of reduction possible", and that reduction "improves prediction" and improves interpretability.

General remarks

Unfortunately, evidence to support one of these claims is not provided (improved prediction performance), and another is subjective (interpretability). Thus, while it is exciting to see this approach (considered a bit of step toward a deep learning approach) employed on this data [please see #6 below], the manuscript requires substantial revision in terms of these claims. The method swap is not proven to be an advance. It is more

accurate to describe it as an equivalent but promising substitution. The work's contribution is purely technical, shows similar performance, and some would find interpretation is more rather than less challenging.

Major points

1. I think it is a matter of opinion that per-measurement labels make for more complicated interpretations and that tensors are fundamentally more interpretable. Language claiming this should be considerably softened and focused more on what the distinct value is to this approach than on its superiority (which is subjective).

Thank you for this suggestion. We have made sure that our language carefully reflects what we think are the benefits of this analysis throughout the manuscript. As discussed in greater detail in point #4, we agree that tensor analysis is not fundamentally easy to interpret but maintain that there is both qualitative and quantitative evidence of its benefits in this application.

2. Good examples of meaningful interpretation of [C]MTF components are given, but the expertise required to do so is not trivial, and the scaling choices complicate matters. It is important to note that these examples represent "just so" stories apparent in the original approach, and that there are also examples that have no known biological basis, as well as those that contradict expectation from prior knowledge that can be found in the data.

We have extensively expanded our pre-processing and normalization of the data, factorization methods, visualization of the resulting factors, rigor in assessment of our prediction models, and discussion of our model interpretations. If the reviewer has remaining concerns, we would appreciate their specific expert input.

3. The authors claim results are "significantly" improved. Fig 5 shows no statistical tests to support this claim, and data that appears to best show equivalence. This claim needs to be proven or eliminated. (A prior version of the manuscript available on biorxiv reports what might be considered to be fundamentally different results in this figure. They appear to have updated their observations but not the language describing them.)

Thank you for pointing out this inconsistency between our language and the analysis results. Indeed, the results changed as we modified both our approach and our reimplementation of Alter *et al*. Specifically, a major difference from an earlier version of the manuscript is we first were reimplementing the Alter *et al* model that used about half of the subject cohort in exchange for including the glycan measurements. We later felt that it would be fairer to include their model with the best prediction performance. Therefore, we now reimplement their model that does not use the glycan measurements but does include the entire cohort.

We have removed this claim of "significant" improvements in prediction performance. We now only claim equivalent accuracy for most predictions (lines 216–225).

4. The distinction between prior approaches and this approach is overstated in some ways. While the authors state that "optimal models required 30-50 variables" in the prior approach, similar performance results from considerably sparser models. There are, however, still fewer features used here, in exchange for some considerable barriers to interpretation.

We have added new analysis regarding the interpretability of the model in Alter *et al*, found in Figure EV2, for comparison to Figure 5. The reviewer is absolutely right that (1) Alter *et al* achieves prediction through a sparse

model and (2) our approach of dimensionality reduction is not inherently intuitive to interpret. However, we maintain there are two important differences to consider regarding the interpretability of each approach.

First, while subjective, we maintain that there is considerable benefit to identifying patterns of covariation in the data. While this is possible with PCA or PLSR, the tensor approach additionally leads to further reduction while separating Fc from antigen effects. We find this extremely helpful. The reviewer rightly notes below that other approaches have been taken to collapse antibody dimensions into summary features. We maintain that this is evidence for the general utility of our approach doing this in a coordinated fashion.

We have added a new supplementary figure, Figure EV2, to demonstrate our second point objectively and quantitatively. While regularized regression methods are extremely effective at variable selection tasks [1], this ability breaks down in the presence of strong correlations among features. Our dimensionality reduction approach is so effective at data reduction *because* these measurements are extremely correlated with one another. This is not a problem—one should expect correlation based on the known biology—but it means that while sparse models are predictive, one cannot attribute significance to the variables selected [2–3]. To show this, we used bootstrapping of the subjects and plotted the variable weights from regularized regression (Figure EV2) [4]. Three bootstrap samples of the ADCD prediction model show very different weights, indicating that one would expect to identify very different predictors given a new, independent dataset. In fact, the presence of IgG1, IgG2, IgG4, and C1q measurements overall drastically changes within each sample. By contrast, bootstrapping with our approach shows consistent weights with small variance relative to the magnitude of the weights (Figure 5). To be clear, this does not indicate that Alter *et al* cannot identify predictive models—these measurements do successfully predict ADCD. However, it shows that one cannot attribute significance to selecting one set of measurements over another. Through data reduction, one can at least assign significance to the selection of certain patterns within the data.

[1]: Candes, E. & Tao, T. The Dantzig selector: Statistical estimation when p is much larger than n . *Ann. Statist.* 35, (2007).

[2]: Efron, B. Prediction, Estimation, and Attribution. *International Statistical Review* 88, (2020).

[3]: Tansey *et al*. The Holdout Randomization Test for Feature Selection in Black Box Models. *arXiv:1811.00645 [stat]* (2021).

[4]: Efron, B. Bootstrap Methods: Another Look at the Jackknife. *Ann. Statist.* 7, (1979).

5. Fv, Fc, and antibody function dimensions have been collapsed intentionally by others (Fv and Fc breadth and polyfunctionality scores, for example). These alternative approaches, which marginally reduce novelty should be mentioned.

Thank you for this suggestion. We have integrated these methods, along with a couple references suggested by reviewer #3, into a paragraph in the introduction covering methods by which others have summarized serology data (lines 83–87). In fact, we think of our approach as a unified method by which to simultaneously perform reduction like this on both dimensions.

6. Why not just go all the way to a deep learning approach?

Indeed, as highlighted in the discussion, we think that further advancements can be made. However, we do not anticipate that a deep learning approach, at least in standard form, would be an optimal data reduction strategy. Methods like variational autoencoders or generative adversarial networks could certainly provide greater dimensionality reduction than CMTF. Supervised neural network models likely can provide accurate predictions of subject classes and immune functional responses. However, interpretation of these models is very limited. Local sensitivity methods exist, but do not provide a “global” picture of the reduction behavior and are extremely susceptible to technical artifacts. More traditional approaches are also generally more effective so long as the structure of the data can be recognized. Given that these are binding events for which mechanistic models can completely account, we expect that non-linear but mechanistic data reduction strategies will eventually be most promising, with the added benefit of integrating prior knowledge. Indeed, we are excited to eventually pursue this by using a multivalent binding model rather than the multiplicative reconstruction of each component.

Even if deep learning models become ideal for modeling these data, recognizing that there is low-rank structure to these data when arranged in tensor form will aid deep learning-based methods. The salient structure of data patterns is essential to application of these methods (e.g., convolution in image analysis). In fact, low-rank tensor structures have had exciting recent integrations with deep learning approaches [1–3].

[1]: Kossaifi et al, Factorized Higher-Order CNNs with an Application to Spatio-Temporal Emotion Estimation, arXiv:1906.06196v2 [cs.LG] 31 Mar 2020

[2]: Bulat et al, Incremental Multi-domain Learning with Network Latent Tensor Factorization, arXiv:1904.06345v2 [cs.CV] 22 Nov 2019

[3]: Kossaifi et al, T-Net: Parametrizing Fully Convolutional Nets with a Single High-Order Tensor, Proceedings of the IEEE/CVF Conference on Computer Vision and Pattern Recognition (CVPR), 2019, pp. 7822-7831

Minor points

1. *Antibody glycan data in this study included antigen-specific antibody glycan profiles, not "antigen-generic" glycan data. The text repeatedly describes this underlying data incorrectly.*

Apologies that we did not describe this well. We meant to convey that, while antibody glycosylation was measured for the gp120-associated fraction, it was not measured separately for each antigen like the other measurements. We have made changes throughout the manuscript to clarify this point.

2. *The introduction states: "To date, no methods have provided a means to holistically visualize the variation in serology measurements." This claim is overly broad and so vague as to be meaningless. It is not clear how standard global visualizations (such as a heatmap) do not meet this standard, and how the method proposed here (which is a dimensionality reduction approach) does accomplish this goal. The claim should be eliminated or properly bounded/specified.*

Thank you for this comment, we have altered our language (lines 87–89) to more exactly point out that previous methods have not taken advantage of the improved dimensionality reduction once measurements are arranged in tensor form according to antigen and Fc features.

3. *What the authors consider to be "missing" data should be clarified. By their definition, this data set might be considered to have infinite missing data, as for example, no T cell data was reported. It feels misleading to*

characterize measurements never assessed as missing. The information they provide is confusing to someone very familiar with this data set, and likely not meaningful to individuals unfamiliar to the data set. Perhaps either greater or lesser detail in the opening section would be an improvement to the current level (which is clarified subsequently).

Sorry for the confusion. We have clarified in the manuscript that most of the “missing” values within the data tensor are created by us enforcing an overall multi-modal structure (lines 166–170). The missing values in the glycan matrix arise because the glycans were only measured for half of the subjects (lines 171–180). The reviewer is correct that missing values can only be defined within a certain analysis structure, but these two situations are qualitatively different from the situation described above describing “infinite” missing data. Even in the previous analysis (Alter *et al*, 2018) the missing glycan measurements meant that separate models had to be constructed either including or excluding the glycan measurements, and the models including them were less predictive.

4. Why does Fig 2a not report variance reconstructed with an alternative method to [C]MTF? What effect does the normalization have on the comparison between PCA and [C]MTF in Fig 2b? Is the normalization to the same value for both PCA and [C]MTF, or does it differ between them?

In Figure 2a, we do not report variance reconstructed with an alternative method because the definition of a component is specific to CMTF. If we were to, for example, report the results of using PCA on this plot, it would provide a misleading representation because each component of PCA is an entirely different construct. We instead compare CMTF and PCA in Figure 2b with factorization size to put these two methods on comparable axes.

Indeed, the normalization is identical between PCA and CMTF. Relevant to this point, please see question 1 from reviewer #3 for a more detailed discussion of normalization. While we were not normalizing previously, we now center the data per-measurement after log-transforming.

5. Figure 3 aims to show accuracy, but reports variance predicted and provides no comparison to alternative approaches.

We have now considerably expanded this section to examine the imputation properties of CMTF more thoroughly and compare it to suitable baselines. In Figure 3a we are not able to use PCA as a baseline because the imputation would be performed for entire columns of the matrix (which is not possible). However, we now compare the variance predicted to the fitting accuracy in Figure 2a (lines 184–186). We did try imputing using the receptor- and antigen-averages (commonly used alternatives), which resulted in Q2X values very close to 0 (lines 188–189). In Figure 3b, we have added evaluation of the imputation accuracy when we leave out individual values instead of entire receptor-antigen pairs across all subjects. This allows us to compare CMTF and PCA in their ability to impute these values.

6. The authors claim that patterns in env and internal antigen binding would be "nearly impossible" to identify without their approach. Antigen-specificity differences are typically readily apparent in humoral response data sets. The authors should prove this assertion or eliminate/soften it.

Agreed, this language was too strong. We now only claim that the pattern was not previously identified using these data (lines 320–338). To help explore how easy it would be to identify this pattern in the absence of our method, we summarized this pattern using just the titer measurements in Figure EV3. In short, just the gp120/p24 ratio can predict controller/progressor status quite well, but not as well as through tensor decomposition. It also should be noted that we only identified the pattern by using CMTF in the first place.

7. Why are randomization results not shown for Fig 5a?

Thank you for this suggestion. We have added this.

8. The supplemental data provides little value.

We have removed unused supplementary plots and added supplementary data based on the reviewer feedback here. There is now closer integration between the main text and the supplemental data. We hope this alleviates the reviewer's concern about its value.

Reviewer #2:

Summary

The manuscript describes a new form of tensor factorization, total tensor-matrix factorization ([C]MTF), with application to systems serology measurements to improve the prediction of immune functional responses, classification of subjects according to their HIV control status and interpretations of models and predictions. [C]MTF capitalizes on the ability to separate the contribution of antigen from immune receptor binding that ultimately increases data reduction by avoiding repetition of the antigens for each receptor measurement. The approach is evaluated with an example systems serology study, previously published by Alter, G. et al which profiled immune responses to predict both functional immune responses and disease status within HIV-infected subjects. The authors show that [C]MTF reduced the dimension (to 2.8%) of the data while preserving about 93% of the original variation. The accuracy of [C]MTF decomposition in predicting functional measurements and subject viral and controller status is shown to be comparable to results published in Alter, G. et al. The authors present an interesting concept to expand on systems serology methods which handle highly dimensional and correlated data. Since the study explains a new approach, the manuscript should refocus the writing to describe specifics of this approach to support the benefits highlighted by the authors.

Major comments:

1. The authors should describe details of the approach for generalized systems serology data. Details on data requirements (i.e., preparation or pre-processing) should be included. Parameter tuning; how is the optimal number of components selected? How is the method evaluated in terms of performance and stability? How is the quality of imputation evaluated?

We have now expanded on each of these points in the results and methods as each issue arises. Specifically, we now cover pre-processing in the methods (lines 436–450), the optimal number of components in the results (lines 210–215), and imputation details in both the results and methods (lines 181–191, 539–550).

We are not sure which aspect of performance and stability the reviewer might be referring to. However, we have considerably expanded the section in the methods describing implementation of the model. As we describe, two methodologic modifications were helpful to resolve the influence of missing values (lines 474–476). Solving is accomplished in a few seconds on a typical modern laptop.

2. The referenced paper for the data (Alter, G. et al) differentiated the subjects across the four groups with approximately 60% accuracy in addition to the prediction of progression and viremia classes. It would be useful to evaluate how well [C]MTF classifies the four groups. It is surprising that Fig. 4a does not show separation of classes in any of the components although the text suggests an improvement in the subject class prediction.

Thank you for this suggestion; we have now added the four-class prediction to Figure 5b.

In inspecting the model decision functions, we came to realize that the model was heavily dependent on the ratios between components, which made it difficult to observe how the model was using the input components. Briefly, PARAFAC models are typically initialized using SVD of the unfolded tensor along each dimension. SVD cannot work with missing values, and so we were filling these in as zero (they were still being handled correctly during the fitting process). To provide a better starting point, we wrote a tailored initialization scheme wherein the completely missing chords were removed from the unfolded tensor prior to SVD. This provided a much better starting point, and more linearly independent subject factors. The differences between classes are now clearer (Figure 4). We have made note of this methodologic improvement in the methods (lines 474–476).

3. What was the rationale for not standardizing the measurements? [C]MTF, like PCA, is a variance maximizing approach; it is expected that measurements with highest variance will dominate the components. This was recognized by the authors for measurements of gp140 HXBc2 (misabeled as HVBc2) and was corrected by multiplying by 0.000001 to scale down this measurement. Not standardizing the data may have contributed to the inability to identify IgG subclass patterns because of FcγR responses overwhelming IgG subclass responses. In Alter, G. et al, IgG subclass assessment revealed that IgG3 contributed to ADCD activity while IgG2 and IgG4 impeded its activity, but [C]MFT did not identify any of the patterns associated with IgG subclass.

We thank the reviewer for bringing up this excellent point about scaling. Indeed, this concern was shared by Reviewer #3. We now standardize the data by centering each measurement after log-transformation, but do not variance-normalize. The choice to omit the latter is because there are many measurements that are essentially zero across many of the subjects and become inflated with variance normalization. All the measurements are on a common unit scale of relative fluorescence, and so the scale of their variance is a meaningful signal.

We have removed any special scaling as we agree that was unjustified. However, we did identify that gp140.HXBc2 and HIV1.Gag were unusual antigens in that they had only one and two receptor measurements, respectively. So little information about these antigens makes their factor values poorly defined. As mentioned in the methods now, we completely remove these two antigens except when reimplementing Alter *et al*.

In Figure EV2, we now show through bootstrapping that the model proposed by Alter *et al* does not have a stable set of variable weights (also see Reviewer #1, question #4). Reconstructing this model leads to distinct model weights with different interpretations. For example, the IgG subclasses vary wildly in their inclusion and

the sign of their effect. Therefore, while the models in this earlier work are predictive, the inclusion or exclusion of any particular variables cannot be taken to be significant for prediction.

4. Is the order of components important? Given that Component 1 describes greater than 75% of the variance, it is surprising that the component is not mentioned anywhere in the text and did not correspond to an important immune pattern. How are the components related? For example, are they orthogonal like in PCA? The description of the data seems to be treating the components entirely as standalone structures.

Thank you for identifying these unclear points. Interestingly, tensor decompositions of Kruskal form (e.g., PARAFAC, CMTF) are different from PCA in (at least) three ways. All three differences stem from the first, which is that the components within each factor are not orthogonal. Components do need to explain different variance along at least two dimensions (otherwise they could be combined) but can be correlated along other dimensions. Sometimes this is referred to by saying that components must be “hyperorthogonal”. Alternative constructions like Tucker decomposition enforce that all components are orthogonal along all factors, but then allow for interactions between components in a way that can be difficult to interpret. We have elected to go with a Kruskal-formatted decomposition because it is much easier to follow the contribution of each component. Because of this, adding another component can influence the composition of the others—two and three component decompositions do not necessarily share two components. Finally, tensor decompositions are not solved in a component-by-component scheme, and so they are not generally ordered in magnitude. However, we have now added a post-decomposition step wherein we order the components by their variance. We also mention their relative scale in the text (lines 245–247).

We realize that these tensor properties are not at all widely known by our intended audience. Therefore, we have added mention of these properties throughout the manuscript where they become relevant.

5. Prediction of ADCD was exclusively explained by Component 5 which is interpreted by the authors to be indicative of the lectin-pathway complement activation, shown by positive weighing of LCA, PNA and SNA responses. In contrast, in Fig. 4b, it appears that Component 5 is dominated by mostly FcγRs rather than LCA, PNA and SNA responses.

This entire section is rewritten with careful checking for consistency with the results.

Minor comments:

1. The writing is overly generalized with background material on systems serology repeated across different sections of the manuscript. It will be useful to reduce background in the abstract (which is currently half of the abstract) and expand on the key aspects of the approach that contributes to systems serology. Background material is again repeated in the results section as the entire first paragraph of "Factor components represent consistent patterns in the HIV humoral immune response". The results section needs to be rewritten as a large fraction of the text corresponds to interpretation or discussion points; this needs to describe the results presented in the figures.

Thank you for this suggestion. We have refined the abstract and results (along with much of the rest of the manuscript, too).

2. The authors should describe the importance of the patterns highlighted in the results with respect to HIV immunity? For example, Gag-binding IgG1 in Component 4 and the shift between surface antigen (gp120, gp140) and p24/p51/p66 binding in Component 5.

We have added a paragraph in the discussion section that connects our findings to HIV immunity (lines 320–338).

3. Fig. 4b shows that the components are dominated by the entire range of FcγR responses (apart from Component 4), so it is not very clear why immune functions are predicted by some components and not others.

In our description of the factorization results we have tried to expand on the unique features of each component. Note that a given component can be distinct along either the receptor or antigen axis. In other words, the variation in a component, which predicts a certain functional response, may indicate variation that is the same as another component for every receptor, but exists to explain a unique pattern of antigen variation. The shared weight across all the FcγR measurements shows how extremely correlated all these measurements are but normalizing the data has helped to reveal more distinct patterns.

4. The authors should support results with data and avoid using the word 'believe'. Component 2 is described to represent broader antigen binding which 'explains the positive correlation with all glycoforms.' These positive correlations are not clearly shown by data in Fig. 4d.

We have rewritten this section entirely, hopefully with clearer explanations.

5. Imputation was tested by removing entire receptor-antigen pairs across all subjects. Could [C]MTF impute measurements missing in only a few subjects? How much missing data could [C]MTF handle while maintaining accuracy?

This is a great question! We have added an analysis for this form of missingness to Figure 3. Please also see reviewer #3, points 3.1, 3.2, and 3.3 for a detailed to response to both these questions.

6. How is imputation evaluated? It seems strange that imputation (Fig. 3) seems to perform much better than original complete data (Fig. 2a).

We believe this was a consequence of only holding out a single random set of values, rather than repeating this process and calculating the average imputation accuracy. We now repeat the imputation process with different sets of receptor-antigen pairs left out and report the average Q2X and variance of this quantity.

7. Remove subscript "a" in Fig. 3

We have added another subplot to Fig. 3 that now makes the subplot label useful.

8. It would be interesting to compare partial least squares (PLS) which also maximizes the covariance between components from two data sets to extract an underlying structure.

Indeed, it would be interesting to compare PLS to the results here, though beyond the scope of our present study. CMTF identifies latent variables that maximize the variance explained across both the glycan and other measurements, while PLS would explain only the variance shared between both data structures. Therefore, this would be more suitable for specifically identifying the relationship between the biophysical profiling and glycans/effector responses. Tensor forms of PLS exist and would have the same benefits as we outline in this study. We have added mention of this point (lines 134, 751). One could also use tensor forms of PLS to make the same functional and class predictions as we have. We chose not to go this route, though, as this would make the resulting factors dependent on the prediction at hand. We wanted to have some view of the molecular measurements themselves, and one set of factors independent of the relevant prediction.

9. What does feature importance quantify in Fig. 5? It is hard to interpret feature importance for class prediction. Does it mean that Component 1 could accurately predict 75% progressors and component 5 could predict 100% of progressors and 100% viremic subjects? Why is there no discrimination of progressors in Component 5 of Fig. 4a?

We now just use linear models for function prediction and classification. As a result, we now simply report the component weights of the models and expect this will be much easier to interpret. As discussed, (lines 474–476 and major point #2), we made some improvements to the factorization and normalization that led to clearer separation of the subject classes. We have also added Figure EV1 which shows the classification boundary using the two most-weighted components visually.

10. CMTF in Fig. 1 is not previously defined.

This has been fixed.

11. Fig. S1 is not referenced anywhere in the text

We added this reference.

12. The word 'Component' is sometimes capitalized and sometimes not capitalized.

We have not been able to identify its inconsistent capitalization. Could you please identify an example? Thank you.

Reviewer #3:

This manuscript develops a form of tensor factorization that incorporates both direct immune interactions (e.g. antibody-virus binding) as well as secondary modes of immune defense mediated by effector cells (e.g. natural killer cells, complement). One key advantage of this method is its ability to incorporate disparate datasets, such as subject-antigen-receptor and subject-glycosylation data, while retaining the underlying structure of each data set. The authors apply this framework to an HIV systems serology study and demonstrate that this vast data set is low-dimensional, permitting ~90% of the variance to be reconstructed using 10 components.

While I found the core concept in this paper very interesting, I was confused about one critical issue: whether the data was centered prior to analysis. If not, then the ~90% variance explained could be a description of how far away the data lies from zero rather than how well the model characterizes the data. As stated below, the authors should clarify this point (both for the [C]MTF and PCA analysis). In addition, there are a few areas of improvement that would help clarify the main results of this manuscript and quantify how robust their results are.

Major comments:

1. Centering the data: Our understanding is that the data used in the factorization algorithm is not centered, and therefore the variance explained by the first factor describes how uncentered the data is instead of how well the model characterizes the data [for a visual example, see <https://i.stack.imgur.com/9vENg.jpg>]. This could artificially inflate their variance explained in Figure 2A and Figure 3, and the authors should rerun their analysis on centered data to see how it affects their analysis.

This is an excellent point, understandably brought up by the other reviewers as well. We now center the data after log-transformation.

1.1 There should be similar clarification for the PCA algorithm used as a point of comparison. Was the data centered before PCA?

To ensure a fair comparison between CMTF and PCA, we normalize the data in an identical way before each algorithm. In fact, the same import and normalization function is used.

1.2 Beyond the issue of centering, the authors should clarify why the PCA results were different from the CMTF results. Was it caused by the fact that in CMTF the eigenvectors do not need to be orthogonal, or did CMTF retain the structure of the antigen-receptor pairs while the PCA analysis lost this structure when the tensor was unfolded?

First, when it comes to the extent of data reduction achieved with PCA versus CMTF, the difference comes from CMTF's ability to "reuse" antigen or receptor patterns across measurements. For example, if a component includes an increase in LCA binding, PCA would still need to represent this increase in the loadings for every LCA-antigen measurement. It is not able to "group" interaction effects across the two dimensions. The difference does not arise through relaxing orthogonality, because tensor decompositions are still "hyper-orthogonal" (see reviewer #2, major point 4), and non-independent components would only reduce the extent of dimensionality reduction. We have expanded our explanation of these distinctions in lines 148–156.

2. Implied structure of the data: The results of the analysis are presented nicely in Figure 4. However, it is unclear what inherent assumptions were made when forcing the data to conform to a simple multiplicative model.

Thank you for pointing out we should justify this implicit assumption about the structure of the data. We have added a discussion to our methods (lines 486–506) justifying a multiplicative interaction between the receptor and antigen dimensions based on the observation that these measurements are produced by a sandwiched binding assay, in which each binding event corresponds to a tensor slice, with shared relative fluorescence units.

2.1 For example, if the subject-receptor-antigen data were converted into different units (e.g., from Molar to $\mu\text{g}/\text{mL}$ units), the framework could perfectly adjust by multiplying all receptor (or antigen) coefficients by the conversion factor. However, if the subject-receptor-antigen data was inverted while not inverting the glycan data (e.g., if binding was reported using association constants K_a rather than dissociation constants $K_D=1/K_a$), then a different set of receptor and antigen values would be inferred. These values could characterize the data much better or much worse, but the authors chose one specific representation for their analysis. Is there some structure to the data that suggests this setup?

Thank you for making this helpful point and something we should have expanded upon in the manuscript. Indeed, non-linear transformations of the data would change the results of the factorization. The biophysical measurements we have included are comparable as they all are fluorescence readings after (1) immobilization based on binding to antigen and (2) detection using a fluorescently tagged reagent. We have emphasized this point on lines 90-95, and in a new methods section justifying the structure of our factorization (lines 486–506). As the glycan measurements do not share this scale, we now explore the effect of adjusting their relative scaling in Figure 2C.

2.2 An additional question is whether the receptor and antigen values shown in Figure 4 represent true biological properties (i.e., whether these inferred values would be the same for other data sets), or if they are dataset specific. For example, if you either: (1) restrict the analysis to different subsets of the data (e.g., 50% of the receptors and 50% of the subjects/receptors/antigens) or (2) remove the glycan data entirely, do you recover similar antigen and receptor values? Ideally, this question would be tested against a separate data set with overlapping antigen-receptor pairs (if one is readily available) but analyzing different subsets of the current data set will suffice.

This is an interesting question! Unfortunately, we have not been able to get ahold of the authors for other studies with similar receptor-antigen measurements (Chung *et al*, *Cell*, 2015). This is frustrating because, as you suggest, ideally, we would show a similar factorization in an independent dataset. However, we have tried to address your question in two ways. First, we performed your second suggestion, removing the glycan data, and saw similar patterns overall. Many components clearly match (here \rightarrow Figure 5: 2 \rightarrow 1, 3 \rightarrow 2, 4 \rightarrow 3, 5 \rightarrow 5, 6 \rightarrow 4). The most obvious change is component 1 here and component 6 in Figure 5, which do not have corresponding matches.

Second, we have now added an independent dataset of SARS-CoV-2 serology and investigated its data reduction (Figure 6). While this does not overlap the HIV dataset, it (1) shows similar benefits of the data reduction method in an independent dataset and (2) as an acute infection, it shows that the factors line up to known serology changes over time, further supporting that these patterns reflect true biological properties.

2.3 "This component also displays negative weighting across gp120 and gp140 antigens, likely reflecting a decrease in antibody titers overall." This is an important point that is worth elaborating upon. How would a difference in antibody titer show up in the data? Wouldn't it decrease the Subject factor in Figure 4A since you would expect all measurements to be lower for that subject?

We would expect that, if a pattern across subjects were variation in antibody titers overall, the component would have equal weighting across antigens, similar weighting across receptors, and varied weights across subjects. The exact makeup of our component weights has changed with our new normalization and initialization improvements, but our explanation of the resulting factors is hopefully now clearer.

2.4 The authors also allude several times to potentially non-linear interactions in the immune system, such as when multiple antibodies interact with one another. Indeed, when analyzing the immune functionalities in Figure 5, the authors use a Gaussian process model to account for non-linearities. Aside from Figure S2, are there other indications of non-linearity in the data? Conversely, are there other pairs of measurements in Figure 4 (analogous to Figure S2) that confirm the linearity assumption?

We have been able to replace the Gaussian process model with linear prediction models through improvements in our factorization initialization and normalization/centering suggested by all three reviewers. As described in our discussion, we do think that these data can reveal non-linear interactions between antibodies. One existing piece of evidence for this is that we do not explain all the variance in the original dataset, and so there are remaining patterns to be identified. However, we think a more mechanistic analysis, separately accounting for the affinity, avidity, abundance, etc, of each antibody species, might provide a more exact route to modeling these data and therefore revealing these interaction effects. Our present analysis is more of a statistical approach to identifying the broad patterns of variation, and as such (1) identifies components that might come about through either interaction effects or variation in a single molecular event, (2) prioritizes the general variation over evidence of specific interactions, and (3) doesn't apply prior knowledge such as IgG-receptor affinities to further constrain the model of the data. Note that another reviewer rightly pointed out that the previous Figure S2 did not provide much value, and so we have removed it from the manuscript.

On the other hand, the best evidence we can provide for the general appropriateness of our linearity assumption is the performance of the data reduction itself. First, CMTF reduces the data more extensively than PCA. Second, CMTF can impute *entire missing chords* from the data based on assuming linear interactions between receptor and antigen effects. We mention both points in a methods section justifying the structure of the model (lines 486–506).

2.5 Is there a hyperparameter to quantify the relative importance of the antigen-receptor vs glycan data? If so, how is its value determined? For example, components #1 and #2 look nearly identical for the glycans in Figure 4D, suggesting that these two components primarily differentiate the antigen-receptor data.

Thank you for identifying this oversight in our analysis. Indeed, the antigen-receptor and glycan data are on differing unit scales, and so their relative scaling is an arbitrary parameter. We now explore this and include a new plot (Figure 2C, explained in lines 157–164) showing the variance explained of each dataset for different relative scaling.

3. Figure 3:

3.1 I was surprised that the variance explained by a single component is so high, especially given the lower (and more slowly growing) R2X in Figure 2A. Is this related to the centering issue, or was it because so few measurements were left out in the Q2X analysis? The authors should clearly state the number of entries (or percentage of entries) that were available for training and the number that were imputed.

Thank you for this suggestion. In the imputation section of our methods, we now state the number of points imputed and the number of points available for training for each of our imputation figures. The imputation accuracy was high in part due to performing the imputation process only once (rectified as suggested right below). Centering the data also affected these results.

3.2 The authors chose 15 receptor-antigen pairs to entirely remove and then reconstruct, but why not run this analysis across all possible receptor-antigen pairs to ensure that those 15 were not special cases? It would also be interesting to remove multiple pairs and see how the reconstruction error changes as the number/percent of missing measurements grows.

Thank you for suggesting this. We have now expanded Figure 3 to include this analysis.

3.3 If this analysis is run multiple times on different subsets of left-out data, it would help to draw bands of uncertainty around each measurement (which may even out the dip in Figure 3 at 8 components). It might also be interesting to expand Figure 3 by drawing multiple curves showing different fractions of left out measurements (e.g., 1%, 5%, 10%, 50%...).

We have now added these bands of uncertainty to Figure 3. Thank you for the suggestion of increasing the fraction of missing values to see how the model behaves. We tried this but found it difficult to draw a clear conclusion—most of the time the factorization would provide accurate imputation up to very high percentages of missing values (~90%), but occasionally would result in extremely negative Q2X values. We also found that the pattern by which values are removed led to very different conclusions. While there is likely more to be learned here, we ultimately decided it was outside the scope of the present study. Both Figure 3A and 3B remove a small number of chords or values to evaluate imputation accuracy in the presence of the current data structure, which is more directly related to the current analysis.

3.4 Could you put the Q2X value in perspective relative to experimental error? For example, when using 1 component, how far off (in absolute units) is the average imputed measurement from the actual value, and how does this correspond to typical measurement error?

Thank you for suggesting we place our variance captured/explained estimate in the context of measurement error. As outlined in Brown *et al*, 2012, these measurements, when they have a large dynamic range, are

reproducible with an r^2 of approximately 0.8 when considered on a log-log axis. Therefore, we are capturing the majority but not quite all the true variation. We have added a mention of this observation (lines 141–144).

4. Figure 5 and 6:

4.1 I am confused by the approach taken towards these functional measurements. The 10 components were derived from the Figure 4 data, so why should combinations of these components explain these additional data of ADCC, ADNP...? Perhaps a different set of components would be better suited to understand these functional measurements using a simple multiplicative model without resorting to a Gaussian process?

We are not entirely sure about the nature of the reviewer's first question. The components were derived from biophysical measurements of the polyclonal antibodies. We then use these measurements to predict functional responses using the polyclonal sera, and the class of subject from which the samples were derived.

Due to an improvement in our factorization code (see initialization scheme and direct fitting in methods), we derived slightly better resulting components. Because of this change we are now able to use linear models for these predictions in Figures 4 and 5. We hope these are easier to interpret.

Finally, the reviewer may also be wondering if one could derive components that are tailor made for predicting certain outputs, akin to PLSR. Indeed, tensor PLS methods exist, though we are not familiar with a tensor PLS method for coupled input data or data with missing values. However, it certainly would be an interesting future direction for exploration.

4.2 Were the "Alter et al" and "Randomized" methods in Figure 5 ever discussed? Was the "Randomized" method even used in Panel A (and the plot markers are hidden behind), or does that method only apply to Panel B?

We apologize for these unclear elements in Figure 5. We have now added the model evaluation with randomized data to Panel A. The text describing this figure has been streamlined based on suggestions here and elsewhere.

=====

Minor comments: [These are all of the take-it-or-leave-it variety]

1. Figure 2B: It was difficult to parse what the "size of factorization" means or interpret why it is written in powers of 2.

1.1 It would help to state that the size of factorization in [C]MTF equals (181 + 22 + 41 + 25) (# of components) and to give a similar statement for PCA.

1.2 Writing the x-axis as powers of 2 seemed arbitrary, and when I first saw this figure, I interpreted the 2^{10} factorization as corresponding to [C]MTF with 10 components.

Size of factorization means the size of the data/degrees of freedom after reduction. We have renamed it to "size of reduced data" to improve clarity. We have also stated the data size after CMTF in the text as well as changed the ticks into numbers without the power form in the figure.

2. Results, First Section:

2.1 When first introducing matrix-tensor factorization, it would help to explicitly state that you are assuming a multiplicative model where each subject, antigen, receptor, and glycan are given some value, and the measurement for a specific subject-antigen-receptor or subject-glycan entry is given by the product of their factors. The first two equations in the Methods do this very well, so consider either bringing those into the main text or describing those equations and pointing the reader to the Methods.

This is a great suggestion. We now point this out in the Figure 1 caption and include a reference to the methods (lines 128–131, 753).

2.2 Even more importantly, you should explicitly describe what the 10 different components represent, since they are so critical to the paper's message. The first time I read the manuscript, I did not understand (and hence ignored) the visual representation in Figure 1C, and the explanation only truly landed when I saw the equations in the Methods.

We have rewritten the results section describing the components to walk the reader through their interpretation more carefully (lines 238–279). This is partly helped by only needing 6 components instead of 10, making a detailed explanation easier.

2.3 Regarding the first two equations in the Methods, explicitly define r as summing over the different components and R as the total number of components (presumably $R=10$ for Figure 4).

Thank you for pointing out this unclarity. We have expanded the method section on tensor factorization to fill in more details and we address this explicitly now.

3. Figure 4:

3.1 Is there an intuitive explanation for why Components 3, 8, and 9 in Panel A are all uniformly purple? Is there a reason why Components 1, 2, 5, 7, and 10 are uniformly green? There is a lot of information in these figures, and it would help to pick out these prominent features that are immediately seen and explain them.

These values can be interpreted just the same as in PCA—within the “direction of variation” for that component, all the subjects point in the positive or negative direction.

We believe this may have been confusing in part due to the sign indeterminacy of Kruskal-formatted tensor decompositions (like CMTF). Like how one can flip the sign of both the scores and loadings in PCA with no effect, any two factors can have flipped signs yielding the same reconstruction. We flip the sign of components so that the subjects and receptor factors maximally point in the positive direction on average. We now explain this when describing the factors (lines 240–249).

The confusion here is probably additionally resolved by our change in normalization, centering, and factorization initialization. This means that the sign of each component can be thought of as a relative change rather than absolute magnitude. Each component is also generally better resolved into positive and negative effects.

3.2 How is the IgG row in Panel B different from the IgG1, IgG2, IgG3, and IgG4 rows?

For each isotype measurement, antigen-specific antibodies were immobilized onto beads, and then the amount of each specific isotype was detected by a PE-conjugated, isotype-specific antibody. "IgG" is a pan-IgG measurement as the detection antibody was instead a PE-conjugated antibody with pan-IgG specificity [1]. We have now added Table 1 to detail how each of these Fc properties were assessed.

[1]: Brown EP, *et al* (2012) High-throughput, multiplexed IgG subclassing of antigen-specific antibodies from clinical samples. *J Immunol Methods* 386: 117–123.

3.3 Do the rows that are nearly completely white (such as the rows from IgG2 to C1q in Panel B) arise because the data for all antigens and subjects against these receptors is zero? For example, I would have expected some signal from the other IgG subclasses (although IgG1 is expected to be the strongest).

This was due to our absolute scaling without centering. After log-transforming and centering the data per-measurement there is a clearer signal from these receptors (Figure 5).

3.4 It would also help to know what the values mean for the antigen-receptor and glycosylation data. Is 0.5 a large value, and if so, what does it represent? Are there units associated with these values, or are they dimensionless fluorescence measurements?

We have added a brief description of the data scale and units at the beginning of our results (lines 240–249). We also now mention in the first results section and methods that all the tensor is made up of dimensionless fluorescence units, while the matrix is in relative abundance units. Due to their different units, we explore their relative scaling in Figure 2C.

4. Results, Fourth Section:

4.1 "and level of natural killer cell activation determined by expression of" → "and the level of natural killer cell activation represented by the expression of". It would also help to state that these are cumulative properties of the entire immune response to distinguish them from the individual factors in Figure 4.

We have made this correction.

4.2 "Separately, we predicted broad subject disease statuses: Controller versus Progressor, and Viremic vs Non-Viremic..." It felt jarring to read about this separate task before hearing about the functional data analysis (discussed in the previous sentence). Also, it would help to explicitly state how this classification is done. Do you learn which components are important by using a subset of subjects and then inferring the classifications for the left-out subjects?

We agree this section (now lines 194–237) was not clearly organized. We have streamlined the text, recognizing that the point of this section is to compare the overall performance of both approaches, not the individual predictions. We also now briefly describe our process of cross-validation.

4.3 "We therefore were able integrate the glycan data" → "We therefore were able to integrate the glycan data"

We have made this edit.

5. Figures 5 and 6:

5.1 It would be clearer if the labels in Panel B were "Viremic/Non-Viremic" and "Controller/Progressor" rather than only listing one.

We have made this adjustment.

5.2 As a minor point, it would be nice if the plot colors and plot markers were more consistent. For example, [C]MTF could be represented by black dots in Figures 2, 3, and 5. It is a small point, but it helps the reader associate the information between plots.

We have made this change so that CMTF and other methods are represented by consistent colors.

6. Results, Sixth Section: "Given the importance of component 6 in predicting nearly all of the functional measurements (Fig. 6A), its importance in predicting viremia is to be expected." This is an odd statement, given that Component 6 was not important for the Progression attribute [whereas according to this sentence, it should have been].

This entire section is rewritten with careful checking for consistency with the results.

7. Discussion:

7.1 "just 8-10 consistent patterns..." Why do you give this range of values?

We have changed this wording.

7.2 "While each functional measurement was predicted through a combination of factors, component 6 contributed to nearly every prediction (Fig. 4B). This suggests that these functions are tuned through both shared and individualized regulatory changes." Is this reference supposed to be for Figure 6B? I do not understand why you are specifically calling out component 6 but no other components. I also don't understand what the second sentence is stating.

We have rewritten this entire section, hopefully in a way that makes each explanation much clearer.

7.3 "While each antigen is treated similarly along one dimension, antigenic sites and strains could be separated into distinct dimensions before decomposition." This is a very interesting point, and it would be worth elaborating upon. Is this possible to do with current measurements, or would it require additional experiments (and if so, how would these experiments separate out the binding contributions to each site)? Both blocking experiments (e.g., Sesterhenn 2019) or mutational experiments (e.g., Angeletti 2017), would be very time- and resource-intensive.

Indeed, the strain-based refinement is likely to be possible with the existing dataset as there are measurements of gp120 and gp140 binding across a variety of strains. The only caveat would be that this would increase the number of missing values in the tensor, since the measurements are not made in a strain-specific manner for the other proteins. However, our method seems to handle a high degree of missing values quite well, and

perhaps the measurements for some of the other HIV proteins could simply be duplicated, given that strains are largely defined and immunologically selected by their surface antigen variation. We expect this would help visualize strain-specific differences, and lead to further data reduction to the extent strain's patterns are shared.

There is some information that should exist in these data already that would help to refine antigen binding patterns within a single protein. First, the patterns across strains provide some information about antigen-specific binding, like your excellent reference (Georgiev *et al*) below. Second, the antibodies that compete for antigen would potentially show a subtly differing pattern compared to antibodies that bind at independent sites. As a simple example, one can only get a 2:1 antibody-to-antigen ratio with the existence of two independent binding interactions. We think that teasing apart these subtleties will require us to integrate a multivalent binding model to model the effects of each binding process more explicitly.

Blocking and mutational experiments would help to refine the details of antigen-specificity, as they could provide direct information about where subsets of antibodies bind. However, there is some flexibility in the scale of these experiments that could help with their resource- and time-intensiveness. For example, if one only blocked or mutated one antigenic site (A), it would still be possible to only distinguish antigen A versus non-antigen A in the data. Measuring a dilution series of polyclonal sera could help to provide more information about antibody competition and affinity. As these assays are multi-plexed across bead-coupled antigens, adding a wider panel of antigens with mutations would just be a matter of cost in constructing the panel, up to hundreds of antigen targets.

We have added some of these points to the discussion section of the manuscript (lines 389–404).

7.4 "Finally, the binding interactions of antibodies, while they produce combinatorial complexity, are a simple set of antigen and receptor binding." As with the previous comment, this is very intriguing. Would you need separate measurements for each monoclonal antibody, or could you decompose measurements from polyclonal serum [like Georgiev 2013]?

This is a fantastic paper! Indeed, we are essentially attempting something very much like this paper with two differences (1) we are operating without the monoclonal antibodies as a standard, and (2) we are separating antibodies based on both their antigen and Fc properties. We expect that with further methodologic refinement and incorporating additional studies to define common Fc features this should absolutely be possible. We have added reference to this paper in our introduction covering previous methods to analyze polyclonal serum (line 87) and future inquiries into strain-specific differences (lines 393–402).

8. Reference 12: The title of this paper is missing "HIV" at the end. The other references should similarly be checked.

We fixed this issue and carefully checked the references.

RE: MSB-2021-10243R, Tensor-structured decomposition improves systems serology analysis

Thank you for sending us your revised manuscript. We have now heard back from the two reviewers who agreed to evaluate your revised study. As you will see below, the reviewers are satisfied with the modifications made and support publication. Reviewer #2 only raises two rather minor points, which we would ask you to address in a minor revision.

We would also ask you to address some remaining editorial issues listed below.

Reviewer #1:

The revised manuscript is significantly improved in clarity, quality, and accuracy, and now also applies the approach to a second data set. The authors should be commended for their thoughtful responses.

Reviewer #2:

The manuscript by Tan et. al. describes a form of tensor factorization, total tensor-matrix factorization ([C]MTF), with application to systems serology measurements to improve the prediction of immune functional responses. The utility of the method is demonstrated on two previously published studies of HIV- and SARS-CoV-2-infected subjects. The accuracy of [C]MTF decomposition in predicting functional measurements and subject viral and controller status is shown to be comparable to results published in Alter, G. et al while the extend of data reduction performed better when compared to PCA.

The concept of [C]MTF is a very interesting addition to current technologies used to characterize the high throughput data of systems serology through data reduction and integration of multiple data sets which helps extract an overall structure to the data. The authors have satisfactorily revised and rewritten the manuscript to address the reviewer's previous concerns. The authors have greatly softened the claim of the superiority of the contribution of the approach, although the claim that CMTF improves the interpretation of the model results is subjective.

Major points: All of the major issues have been adequately addressed

Minor points:

1. Describe each subplot of Figure EV2 in the legend.
2. Change 'very slightly better than a random forest classifier in previous analysis' in lines 305-306 to 'comparable to a random forest classifier in previous analysis' given that the confidence intervals likely overlap.

Point-by-point response to reviewers

Reviewer #1:

The revised manuscript is significantly improved in clarity, quality, and accuracy, and now also applies the approach to a second data set. The authors should be commended for their thoughtful responses.

Reviewer #2:

The manuscript by Tan et. al. describes a form of tensor factorization, total tensor-matrix factorization ([C]MTF), with application to systems serology measurements to improve the prediction of immune functional responses. The utility of the method is demonstrated on two previously published studies of HIV- and SARS-CoV-2-infected subjects. The accuracy of [C]MTF decomposition in predicting functional measurements and subject viral and controller status is shown to be comparable to results published in Alter, G. et al while the extend of data reduction performed better when compared to PCA.

The concept of [C]MTF is a very interesting addition to current technologies used to characterize the high throughput data of systems serology through data reduction and integration of multiple data sets which helps extract an overall structure to the data. The authors have satisfactorily revised and rewritten the manuscript to address the reviewer's previous concerns. The authors have greatly softened the claim of the superiority of the contribution of the approach, although the claim that CMTF improves the interpretation of the model results is subjective.

Major points: All the major issues have been adequately addressed

Minor points:

1. Describe each subplot of Figure EV2 in the legend.

We have added this (lines 832–835).

2. Change 'very slightly better than a random forest classifier in previous analysis' in lines 305-306 to 'comparable to a random forest classifier in previous analysis' given that the confidence intervals likely overlap.

Done.

RE: MSB-2021-10243RR, Tensor-structured decomposition improves systems serology analysis

Thank you again for sending us your revised manuscript. We are now satisfied with the modifications made and I am pleased to inform you that your paper has been accepted for publication.

Corresponding Author Name: Aaron Meyer

Manuscript Number: MSB-2021-10243